


**Emission of biogenic volatile organic compounds from warm and oligotrophic**
**seawater at the Eastern Mediterranean**
Chen Dayan[1], Erick Fredj[2*], Pawel K. Misztal[3], Maor Gabay[1], Alex B. Guenther[4],
Eran Tas[1*]
[1]Hebrew University of Jerusalem, Rehovot, Israel (chenescohen@gmail.com), [2]Lev
Academic Center, Jerusalem, Israel, [3]The University of Texas at Austin, Austin,
Texas, USA, [4]University of Irvine California, Irvine, California, USA
* Correspondence to:
Eran Tas, The Department of Soil and Water Sciences, The Robert H. Smith Faculty of
Agriculture, Food and Environment, Hebrew University of Jerusalem, Rehovot, Israel.
eran.tas@mail.huji.ac.il,
Erick Fredj, Department of Computer Science, Jerusalem College of Technology, Jerusalem,
Israel, erick.fredj@gmail.com



**Abstract**
Biogenic volatile organic compounds (BVOCs) from terrestrial vegetation and marine
organisms contribute to photochemical pollution and affect the radiation budget, cloud
properties and precipitation via secondary organic aerosol formation. Their emission
from both marine and terrestrial ecosystems is substantially affected by climate
change in ways that are currently not well characterized. The Eastern Mediterranean
Sea was identified as a climate change "hot spot", making it a natural laboratory for
investigating the impact of climate change on BVOC emission from both terrestrial
and marine vegetation. We quantified the mixing ratios of a suite of volatile organic
compounds (VOCs), including isoprene, dimethyl sulfide (DMS), acetone,
acetaldehyde and monoterpenes, at a mixed vegetation site ~4km from the
southeastern tip of the Levantine Basin, where the sea surface temperature maximizes
and ultra-oligotrophic conditions prevail. The measurements were performed between
July and October, 2015, using a proton-transfer-reaction–time-of-flight mass
spectrometer. The analyses were supported by the Model of Emissions of Gases and
Aerosols from Nature (MEGAN 2.1). For isoprene and DMS mixing ratios, we
identified a dominant contribution from the seawater. Our analyses further suggest a
major contribution at least for monoterpenes from the seawater. Our results indicate
that the Levantine Basin greatly contributes to isoprene emission, corresponding with
mixing ratios of up to ~9 ppbv several km inland from the sea shore. This highlights
the need to update air-quality and climate models to account for the impact of sea
surface temperature (SST) on marine isoprene emission. The DMS mixing ratios were
one-to-two orders of magnitude lower than those measured in 1995 in the same area,
suggesting a dramatic decrease in emission due to changes in the species composition
induced by the rise in SST.





1. **Introduction**
Biogenic volatile organic compounds (BVOCs) emitted from terrestrial vegetation
and marine organisms significantly affect air pollution and health via increasing
regional photochemical $O_3$ pollution (Curci et al., 2009), enhancing local $O_3$ removal
via chemical reaction (Calfapietra et al., 2013) and serving as precursors for
secondary organic aerosol (SOA) formation (Griffin et al., 1999; Lang-Yona et al.,
2010; Ren et al., 2017). Considering the large global emission rate of BVOCs, mostly
from terrestrial vegetation (700–1000 TgC year$^{-1}$; Laothawornkitkul et al., 2009),
biogenic SOA formation further impacts the radiation budget, precipitation, and
climate (Chiemchaisri et al., 2001; Wuebbles et al., 1989). BVOC oxidation likewise
increases $CO_2$ levels, as a direct product, and methane concentrations, by reducing the
oxidation capacity (Penuelas et al., 2010).
Only a minor fraction of all BVOCs (>10,000) have sufficient reactivity and
emissions to play an important role in the climate and photochemistry (Guenther,
2002). Here, we focus on some of the important emitted reactive BVOCs, including 2-
methyl-1,3-butadiene (isoprene), dimethyl sulfide (DMS), and some oxygenated
VOCs (OVOCs). Emission of isoprene from vegetation has received a lot of attention
in recent years, because this compound has the highest global emission rates among
all reactive BVOCs from vegetative sources (Guenther, 2002), and due to its high
photochemical reactivity and contribution to SOA amounts, estimated to be at least
27-48% of total global SOA formation (Carlton et al., 2009; Meskhidze and Nenes,
2007). It is also well recognized that isoprene is emitted from seawater, too (Bonsang
et al., 1992; Goldstein and Galbally, 2007; Kameyama et al., 2014; Liakakou et al.,
2007; Matsunaga et al., 2002), by marine organisms, including phytoplankton,
seaweeds and microorganisms (Alvarez et al., 2009; Broadgate et al., 2004;
Kameyama et al., 2014; Kuzma et al., 1995). Although the emission rates of isoprene
into the marine boundary layer (MBL) are substantially smaller than terrestrial
emissions, 0.1–1.9 TgC year$^{-1}$ (Arnold et al., 2009; Palmer and Shaw, 2005) vs. 400–
750 TgC year$^{-1}$ (Arneth et al., 2008; Guenther et al., 2006, 2012), they play an
important role in SOA formation (Hu et al., 2013) and photochemistry (Liakakou et
al., 2007) in the marine environment, particularly in more remote areas (Ayers et al.,
1997; Carslaw et al., 2000).
Dimethyl sulfide (DMS) is another important source for SOA formation and
for atmospheric sulfur. The DMS emission rate is much higher from seawater than
from terrestrial vegetation, because the marine environment contains different types of
phytoplankton, algae, and microbial activity (Gage et al., 1997; Stefels et al., 2007;
Vogt and Liss, 2009). DMS emission in the MBL is estimated at 15-34.4 Tg year$^{-1}$
(Kettle and Andreae, 2000; Lana et al., 2011), the largest natural source of sulfur in
the atmosphere (Andreae, 1990; Simo, 2001), accounting for nearly half the total
sulfur emission to the atmosphere (Dani and Loreto, 2017).
OVOCs, including aldehydes, alcohols, ketones and carboxylic acids, can
induce tropospheric $O_3$ formation via $RO_2$ formation (Monks et al., 2015; Müller and
Brasseur, 1999; Singh, 2004) and act as OH precursors, particularly in the upper
troposphere (Lary and Shallcross, 2000; Singh et al., 1995; Wennberg et al., 1998).
Similarly to isoprene and DMS, OVOCs serve as precursors to SOA formation
(Blando and Turpin, 2000).
Emission of BVOCs from both terrestrial and marine sources is fundamentally
influenced by climate changes. For instance, most BVOC emissions from terrestrial
vegetation tend to increase exponentially with temperature (T) (Goldstein et al., 2004;
Guenther et al., 1995; Monson et al., 1992; Niinemets et al., 2004; Tingey et al.,



1990), while drought can negate the effect of temperature on the emission rate from
vegetation (Holopainen and Gershenzon, 2010; Llusia et al., 2015; Peñuelas and
Staudt, 2010; Schade et al., 1999). Seawater acidification and sea surface temperature
(SST) increases significantly affect BVOCs in various ways, including by altering the
biodiversity, spatial and temporal distribution and physiological activity of marine
organisms, influences that are currently not well characterized (Beaugrand et al.,
2008, 2010; Bijma et al., 2013; Bopp et al., 2013; Dani and Loreto, 2017).
Accordingly, the effect of climate change on BVOC emissions into the MBL is
largely unknown (Boyce et al., 2010; Dani and Loreto, 2017).

The Eastern Mediterranean Basin has been recognized as a highly responsive

region to climate change and has been aptly named a primary "climate change
hotspot" (Giorgi, 2006; IPCC, 2007; Lelieveld et al., 2012). Being both warm and
oligotrophic, it gives rise to the dominance of unicellular and small plankton such as
cyanobacteria (Krom et al., 2010; Rasconi et al., 2015), making it an attractive site to
study the impact of anthropogenic stress and climate change on marine BVOC
emissions.

At the southeastern tip of the Mediterranean Basin is the Levantine Basin,

which is ultra-oligotrophic and the warmest region in the Mediterranean Sea (Shaltout
and Omstedt, 2014; Azov, 1986; Krom et al., 2010; Psarra et al., 2000; Sisma-Ventura
et al., 2017; Yacobi et al., 1995), particularly in its northern section (Efrati et al.,
2013; Koç et al., 2010). This region has experienced a significant increase in SST
during the last decade ($+0.12\pm0.07$ °C year$^{-1}$ (Ozer et al., 2016), with temperatures
exceeding 30°C 2 km from the coastline in 2015 (IOLR, 2015).

Most of the surface BVOC measurements at the Eastern Mediterranean are

from Finokalia, Crete (Kouvarakis and Mihalopoulos, 2002; Liakakou et al., 2007).





To the best of our knowledge, only a few measurements of BVOCs were performed in
the Levantine Basin, including BVOC emissions in Cyprus (e.g., Debevec et al.,
2017; Derstroff et al., 2017)  and DMS measurements in Israel (Ganor et al., 2000).

This study includes the first measurements of a suite of BVOCs near the

Levantine Basin coast. The measurements were performed in a mixed-Mediterranean
vegetation shrubbery, with the main objective of studying the contribution of both
seawater and local vegetation to the concentrations of key BVOCs, including
isoprene, DMS, acetone, acetaldehyde and monoterpenes (MTs). A special focus was
given to the effect of meteorological conditions on the contribution of each source to
the measured concentrations.


**2. Methods**
2.1 *Measurement site*
Field measurements were performed in Ramat Hanadiv Nature Park (33°33'19.87"N,
32°56'50.25"E). The measurement site is situated at the edge of the park's memorial
garden. This site is located about 3.6 km from the Mediterranean shore, 120 m above
sea level. The characteristics of the park are described in detail by Li et al. (2018) and
briefly in Fig. 1. The nature park consists of mixed natural Mediterranean vegetation:
*Quercus calliprinos* (~25%), *Pistacia lentiscus* (~20%), the sclerophyll *Phillyrea*
*latifolia* (broad-leaved phillyrea) (~7.5%), invasive species (~10%), *Cupressus* (5%),
*Sarcopoterium spinosum* (~2%), *Rhamnus lycioides* (~2%) and *Calicotome villosa*
(~1%). The park's western part features a few scattered *Pinus halepensis* (<5%)
combined with planted pine (*Pinus halepensis* and *Pinus brutia*) and cypress
(Massada et al., 2012). During the measurements, the average canopy height was


~4.5 m, the leaf area index was ~1.3 and the vegetation cover fraction was ~0.5. The
site is exposed to various anthropogenic contributions: Two highways are located
1.5 km and 2.5 km west of the measurement site, a power plant ("Hadera") is at a
distance of 11 km south of site, and a major industrial zone (Haifa) is 30 km to the
north. Aquaculture farms totaling ~6 km in length, located 3.2 km to the west of the
site, could potentially also contribute to BVOCs at the site.

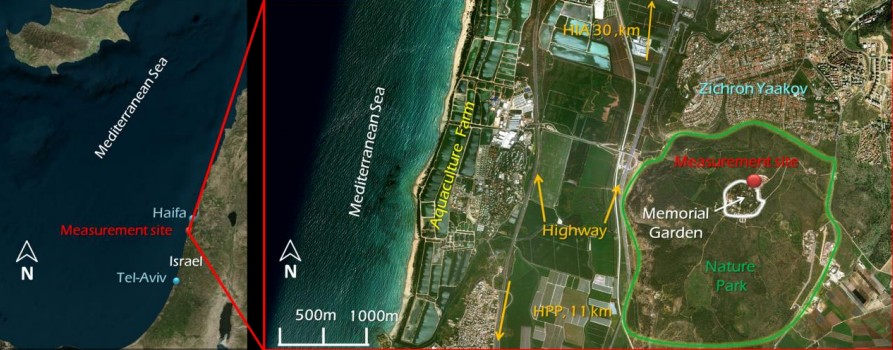

**Figure 1. Satellite images of the measurement site at Ramat Hanadiv Nature Park.** Left: Location
of the measurement site (red dot). Right: Zoom-in on the surrounding area of the measurement site (red
dot). Background imagery from ©Google Earth.

2.2 *Field Measurements*
The field measurements were taken at the Ramat Handiv site from the summer until
the late autumn of 2015 (July 6–October 12, 2015).  The set of instruments included a
platform for eddy covariance measurements of BVOCs, $O_3$, carbon dioxide ($CO_2$) and
water vapor ($H_2O$), trace-gases mixing ratios, including $O_3$, $NO_X$, $SO_2$ and CO, and
basic meteorological conditions, using an air-conditioned mobile laboratory and two
towers (Fig. S2). The sampling routine and schematic of the setup were described in
(Li et al., 2018) and are summarized in Fig. S2.



**Measurement and analysis of VOC concentrations:** VOC measurements
were conducted using a proton-transfer-reaction time-of-flight mass spectrometer
(PTR-ToF-MS 8000 Ionicon Analytik GmbH, Innsbruck, Austria). A detailed
description of the instrument can be found in: Graus et al., 2010; Jordan et al., 2009.
The PTR-ToF-MS was placed inside an air-conditioned mobile laboratory, and
ambient air was pulled at a rate of about 35 l min$^{-1}$ through an external PFA Teflon
tube (3/8" OD, 5/16" ID) and subsampled by the PTR-ToF-MS at a rate of 0.5 l min$^{-1}$
via a 1/16" OD (1 mm ID) polyetheretherketone (PEEK) tube. The instrument inlet
and drift-tube were heated to 80ºC, the drift pressure was set to 2.3 mbar, and the
voltage to 600 V; all the settings were maintained at constant levels throughout the
measurements, corresponding to the E/N ratio of 140 Td.
The PTR-ToF-MS raw hdf5 (h5) files were preprocessed by a set of routines
included in the *ptrwid* processing suite within an Interactive Data Language (IDL)
environment and described in detail in Holzinger, 2015. Further data processing was
performed by customized multi-step Matlab (Mathworks Inc.) postprocessing
routines, which included processing of calibrations, zero air, and ambient
measurements, chemical formula assignment, and comprehensive quality control
similar to Tang et al., 2016. The list of compounds inferred from chemical formulas
and further analysis (e.g., correlation matrix, diel variability, and fragmentation
patterns) is shown in Table S2. The uncertainties are listed according to whether a
compound was explicitly calibrated, an accurate proton reaction rate constant was
used (Sect. S1; Cappellin et al., 2012; Yuan et al., 2017), or a default reaction rate
constant ($2.5 \times 10^{-9}$ cm sec$^{-1}$) for unidentified ions was employed (not reported here).
**Measurements of other trace gases and micrometeorology**: Complementary
measurements included the quantification of mixing ratios of carbon monoxide (CO),



sulfur dioxide ($SO_2$), nitrogen oxides ($NO_x \equiv NO + NO_2$) and ozone ($O_3$), using
models 48i, 43s, 42i and 49i, respectively (Thermo Environmental Instruments Inc.,
Waltham, MA, USA), with manufacturer-reported limits of detection of 4.0 ppm, 0.1
ppbv, 0.4 ppbv and 1.0 ppbv, respectively. These monitors were periodically
calibrated to avoid drift in their accuracy. Trace-gas mixing ratios were recorded by a
CR1000 data logger at a frequency of 1 min. Wind speed and wind direction were
measured using an R. M. Young Wind Monitor 05103 (R.M. Young, Traverse City,
MI, USA), the air temperature and relative humidity with a Campbell CS500 probe
(Campbell Scientific, Logan, UT, USA), and the global radiation with a Kipp &
Zonen CM3 Pyranometer (Kipp & Zonen, Delft, Netherlands). The measured data
were recorded by a CR10X data logger (Campbell Scientific) at 10 Hz frequency.
Overall, the measurements resulted in 20 days of high-quality, complete data, which
were divided into six different periods due to instrument downtime (see Sect. S1).

2.3 *Model simulations of BVOC emission*
The Model of Emissions of Gasses and Aerosols from Nature version 2.1
(MEGANv2.1; Guenther et al., 2012) was applied to estimate the emission flux of
BVOCs from the nature park, according to the vegetation type, and the on-site
measured solar radiation, temperature, soil moisture, vegetation-cover fraction and
leaf area index, using the following general formula to estimate the emission flux of
species i ($F_i$):
$$F_i = \gamma_i \sum \varepsilon_{i,j} \chi_j \,,$$
where   $\varepsilon_{i,j,}$  is the emission factor (representing the emission under standard
conditions) of vegetation type j, $\gamma_i$ is the emission activity factor, which reflects the
impact of environmental factors and phenology, and $\chi_j$ represents the vegetation



effective fractional coverage area. The landscape average emission factor was
estimated using the observed plant species composition at the field site (see Sect. 2.1).
The major driving variables of the model are solar radiation, calculated leaf
temperature, leaf age, soil moisture, and leaf area index. The actual measured
parameters at Ramat Hanadiv were used as input to the model, including vegetation
and soil type, vegetation coverage, fraction and leaf area index, soil water content and
*in situ*-measured meteorological data. Note that only the nature park was simulated by
MEGANv2.1, while potential emissions from a nearby, relatively small "Memorial
Garden" were not taken into account.


**3. Results and discussion**
3.1 *Seasonal and diel trends in measured BVOCs*
Figure 3 presents the daytime average mixing ratios for selected VOCs measured in
the field, along with the corresponding daytime average temperature. The presented
data are not continuous, due to instrument unavailability, and were, therefore,
separated into seven different measurement periods during the year 2015, as shown in
Table 1.





**Table 1**. Measurement periods and corresponding daytime mean of meteorological parameters used for
the analyses*

| Day of Year (calendric day) | T (°C) | PAR (W/m²) | RH (%) | WDD (°) | WDS (m/s) |
|---|---|---|---|---|---|
| 187-188 (6-7 July) | 26.5 | 522.0 | 69.2 | 283.2 | 3.3 |
| 197-199 (16-18 July) | 27.7 | 477.1 | 73.2 | 251.2 | 3.2 |
| 205-207 (24-26 July) | 29.6 | 533.4 | 66.6 | 319.6 | 4.4 |
| 225-226 (13-14 August) | 29.6 | 481.6 | 65.3 | 288.2 | 3.0 |
| 257-260 (14-17 September)** | 29.7 | 395.4 | 69.1 | 320.6 | 3.9 |
| 268-269 (25-26 September) | 29.3 | 461.0 | 56.0 | 324.2 | 3.7 |
| 282-285 (9-12 October) | 28.6 | 397.7 | 53.1 | 329.3 | 3.5 |


* See Table S1 for data availability and exclusion.
** Discussed only in relation to Fig. 3 considering irregular meteorological conditions.

Fig. 3 presents both VOCs dominated by biogenic sources (BVOCs) and VOCs
dominated by anthropogenic emission sources (AVOCs), although no compound can be
regarded as exclusively biogenic or anthropogenic. The former include monoterpenes
(MTs; m/z=137.133, m/z=95.086, m/z=81.070), isoprene+2-Methyl-3-buten-2-ol (MBO)
(m/z=69.071) dimethyl sulfide (DMS; m/z=63.062), acetone (m/z=59.049), acetaldehyde
(m/z=45.033) and the sum of methyl vinyl ketone and methacrolein (MVK+MACR;
m/z=71.048)(Janson and de Serves, 2001; Kanda et al., 1995; Karl et al., 2003; Park et
al., 2013b). The latter include 1,3-butadiene (m/z=55.055)(Filipiak et al., 2013) and
hydrogen sulfide ($H_2SH^+$;m/z=34.995)(Li et al., 2014). It is interesting to note that both
MVK and MACR can have an anthropogenic source and be an oxidation product of
isoprene (Fares et al., 2015; Jardine et al., 2013). Furthermore, this signal may
correspond to 2,3 dihydrofuran.





The dominating source behavior for BVOCs is reflected in their diurnal cycle, which was
characterized by an increase in their mixing ratios from morning to around noontime or
afternoon, followed by a gradual decrease until sunset (see Figs. S5-S9). We found
similar day-to-day trends in the mixing ratios of all BVOCs, particularly of acetone,
acetaldehyde and the MTs. This strongly reinforces the predominantly biogenic origin
for these four species, considering that MTs are expected to be primarily emitted from
biogenic sources in the studied area, in the absence of any nearby wood industry. $H_2S$
and butadiene show significantly different trends in the mixing ratios, suggesting a
dominating anthropogenic contribution for these species, with a potential contribution
from microbial activity (Misztal et al., 2018).
Overall, the day-to-day trend in the BVOC mixing ratios appears to follow the
temperature, but exhibits only a relatively weak correlation with daily temperature
variation (Fig. 3). DMS showed the strongest correlation with the average daytime
temperature ($r^2$=0.27; see Sect. 3.2.2), corresponding to a significant increase in the
mixing ratios between early summer (DOY=188) and the end of summer (DOY=254),
which decreased during autumn (DOY=255 to DOY=283). The other BVOCs, except for
isoprene+MBO, showed a gradual increase in their mixing ratios during summer (DOY
197-260), and a decrease during autumn (DOY 268-285), which can be explained by the
correlation with T (Fig. 3). We attribute the extreme mixing ratios during DOY 257-260
to extreme meteorological conditions; this period was characterized by high wind speeds
and relatively low solar radiation, which can facilitate a shallow boundary layer and, in
turn, higher VOC mixing ratios (see Sect. S4).
While the diurnal profile of isoprene+MBO suggests a predominantly biogenic source
(Fig. 5), its day-to-day mixing ratios showed higher variability, which was quite different
from both DMS and the other BVOCs.

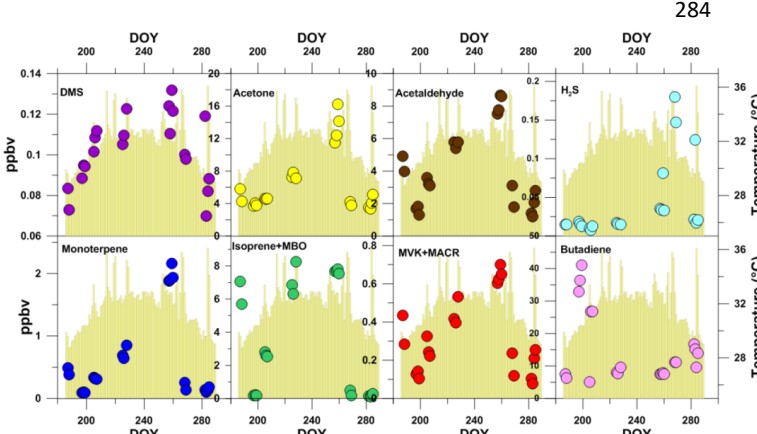

**Figure 3.** The daytime average of selected VOCs. Yellow bars indicate the average daily temperature.

DOY indicates the day of year. For average diurnal profiles, see Fig. S5-S9.

3.2 *Origin of the BVOCs*

To explore the potential sources of the BVOCs, we calculated for each wind sector the percentage of time corresponding with several mixing-ratio ranges, individually for each species (Fig. 4). Our findings indicate elevated mixing ratios for westerly and southeast wind components. The relatively elevated mixing ratios from the southeast can be attributed to emissions from the memorial garden, where frequent thinning of the vegetation can contribute to the generally elevated mixing ratios of plant-wounding BVOCs such as hexenal and hexanal (e.g., Brilli et al., 2011; Ormeño et al., 2011; Portillo-Estrada et al., 2015) from this direction. The elevated mixing ratios from the west may point to an additional contribution from marine origin, such as the Mediterranean Sea and/or the aquafarms, considering that the measurement site is surrounded by nearly homogeneous vegetation in all directions except for the memorial garden (Fig. 1). We found a smaller relative contribution of DMS from the southeast compared to the other BVOCs. The MEGANv2.1 simulations suggested no significant emission of isoprene from the nature park; the relatively strong



contribution of isoprene+MBO from the southeast can be attributed to MBO
emissions from conifer trees (Gray et al., 2003) in the memorial garden. Similar
trends in the day-to-day variation of MVK+MACR, isoprene oxidation products, and
isoprene+MBO (Fig. 3) could imply the contribution of the memorial garden to
isoprene emission, but this possibility is ruled out by kinetic analysis (see Sect. S2).
The elevated mixing ratios of isoprene+MBO from the west may be attributed to the
emission of isoprene from marine organisms, as discussed in Sect. 3.2.1. The origin of
DMS is further addressed in Sect. 3.2.2.

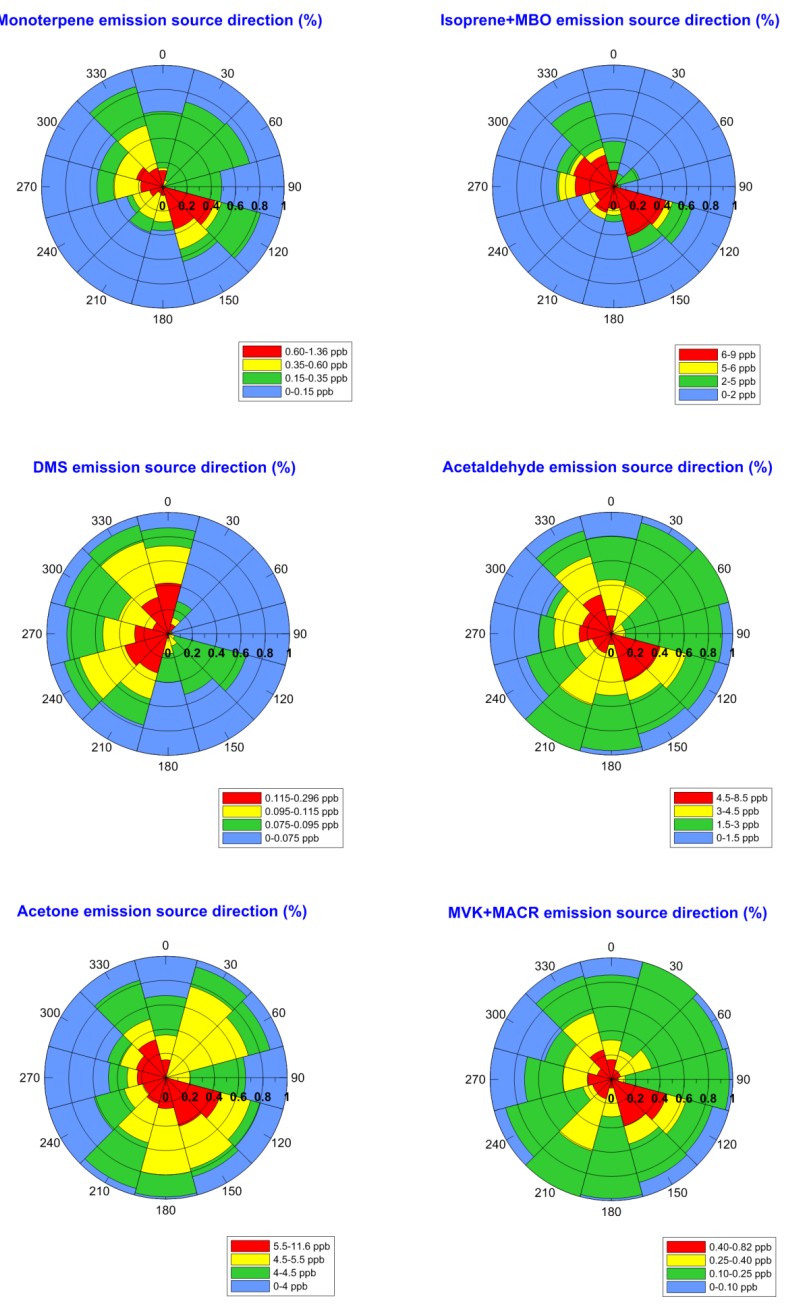


**Figure 4**. BVOC mixing ratios as a function of the contribution from each wind sector. The radial

dimension represents the fraction of time for each wind sector, for which the mixing ratios were within

a certain range, as specified in the legend.





*3.2.1 Origin apportionment of measured Isoprene+MBO*
As demonstrated in Sect. 3.1, the isoprene+MBO day-to-day variations differed from
those of most of the BVOCs, with remarkably high variations in its mixing ratios,
ranging from 0.03 ppbv to nearly 9 ppbv (Fig. 3),  while the seasonal variation in its
mixing ratios did not correlate with temperature (see Fig. S12). The low correlation
between the diurnal profile of isoprene and carbon monoxide (see Fig. S10) strongly
supports no significant contribution to isoprene mixing ratios from traffic on the two
highways to the west (Fig. 1), considering that CO can be used as an indicator for
incomplete combustion of fossil fuels. The Positive Matrix Factorization model
(PMF) further predicted that isoprene+MBO has a common source with other BVOCs
and not with AVOCs (see Fig. S3).

Figure. 5 presents the regressions of isoprene+MBO mixing ratios vs. T for the

six measurement periods. For the two periods with high and low isoprene+MBO
mixing ratios, there was a clear typical biogenic diurnal trend, with a maximum
around noontime. This finding reinforces the notion that isoprene+MBO originates
predominantly from biogenic sources. We did not, however, observe a positive
correlation between isoprene+MBO mixing ratios and air T in all six periods (Table
1). Furthermore, in most cases, we found no exponential increase in isoprene+MBO
with air T, as is expected in the case of a nearby local biogenic source (e.g., Bouvier-
Brown et al., 2009; Fares et al., 2009, 2010, 2012; Goldstein et al., 2004; Guenther et
al., 1993; Kurpius and Goldstein, 2003; Richards et al., 2013). This might be related
to the fact that the m69 signal is affected by the mixing ratios of both isoprene and
MBO emitted locally and further away, while the local air temperature did not reflect
changes of more distant leaf temperatures or SSTs.




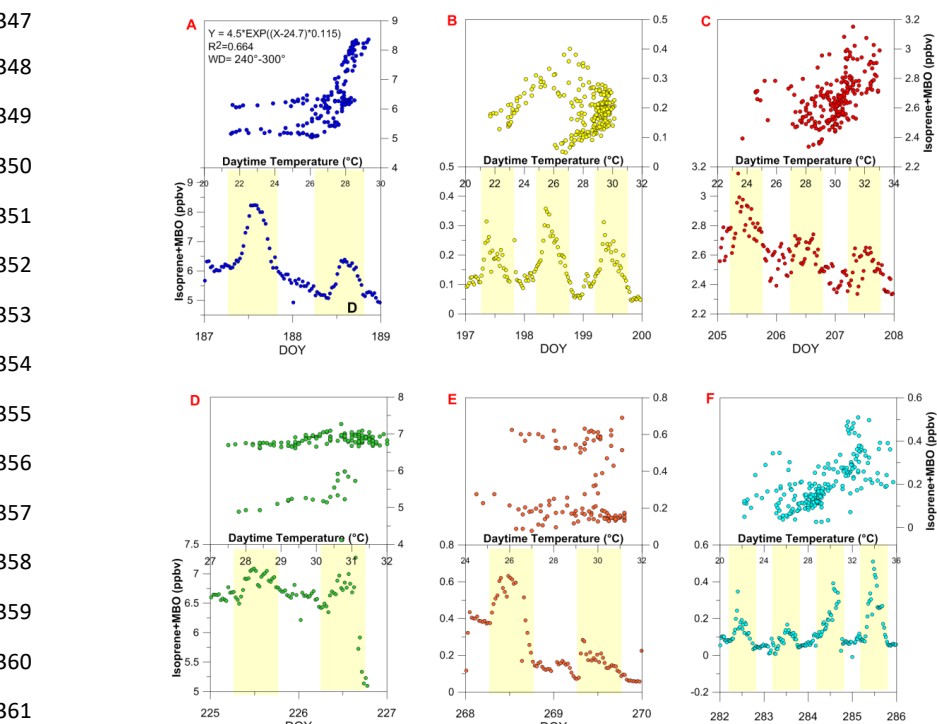

**Figure 5.** Isoprene+MBO (m69) diurnal average mixing ratios and time series. (A-F) The regression between the measured MBO+isoprene (ISP+MBO) and T (upper panels) and the time series of isoprene+MBO (lower panels) for the six measurement periods: DOY 187-188 (A), DOY 197-199 (B), DOY 205-207 (C), DOY 225-226 (D), DOY 268-269 (E), DOY 282-285 (F). The regression between the measured MBO+isoprene and T (upper panels) excludes measurements associated with wind direction from the memorial garden (90°-150°).

We used the fact that MBO can be also detected at m/z=87.0805 (m87), which typically accounts for 13-25% of the total MBO signal (Kaser et al., 2013; Park et al., 2013a, 2012, 2013b) to learn about the ratio between the isoprene and MBO mixing ratios. Figure 6a presents the mixing ratios for m69 vs. m87/m69. Periods with high mixing ratios for m69 were associated with a very low m87/m69 ratio (less than 2%), which suggests that the emissions are predominantly of isoprene. Fig. 6a indicates also that m87/m69>25% was mostly measured during nighttime, twilight and early





morning. For low m69, the ratio matches the MBO typical ratio, m87/m69, which
ranges between 13-25% or higher (Fig. 6a). Furthermore, in those relatively low m69
mixing ratio periods, the ratio between the measured m69 and the MT relative signals
match the [MBO]/[MT] from MEGANv2.1. This reinforces the hypothesis that high
isoprene+MBO mixing ratios predominantly result from isoprene emission, whereas
low mixing ratios are primarily from local vegetation MBO emissions.
Figure 7 further presents the diurnal profile for m87/m69<13%, as well as the
corresponding mixing ratios versus T, separately for each measurement day.
Interestingly, some of the measurement days presented in Fig. 5 were associated with
no m87/m69<13%, which is why there are fewer measurement days in Fig. 7 than in
Fig. 5. The diurnal profiles in Fig. 7 support a biogenic origin for isoprene, although
they were more scattered for 25-27 of July. Fig. 7 also demonstrates the positive
correlation between the isoprene mixing ratio and T during all measurement days,
while in several days a sharp increase in isoprene with T occurred for T>~26-28°C
(e.g., 6,7 July and 16 August). In general, a higher correlation with temperature was
obtained for m87/m69<13% (Fig. 7) than for all m69 signals (i.e., Fig. 7 vs. Fig. 5),
reinforcing the biogenic origin for isoprene with a relatively strong dependency on T.
Furthermore, regression of m87/m69>13% with T does not indicate a clear
dependency of mixing ratios on T, suggesting different emission controls for the
MBO and isoprene (see Fig. S4). The MBO mixing ratios tended to be controlled by
both T and solar radiation, while isoprene was predominantly governed by the former,
in agreement with a previous study (see Kaser et al., 2013).
To study the origin of isoprene, we analyzed the fraction of time for which
m87/m69<13% vs. wind direction (Fig. 6b). We found that m87/m69<13%
predominantly corresponds with a western origin. These results suggest a significant



contribution of isoprene from the sea or the aquaculture farm located at the,
considering that the measurement site is nearly homogeneously surrounded by mixed-
Mediterranean vegetation, except for the memorial garden to the southeast.
Furthermore, MEGANv2.1 simulations predicted a negligible emission rate for
isoprene from the nature park.
In some cases (~4% of the time), elevated m87/m69>13% was recorded also
from the southwest and northwest, which according to simulations by HYSPLIT can
be entirely attributed to transport from either the sea or the aquaculture farms (see Fig.
S11). The relatively small fraction of time for which m87/m69<13% is from the
southeast can be attributed to the emission of MBO from conifers.

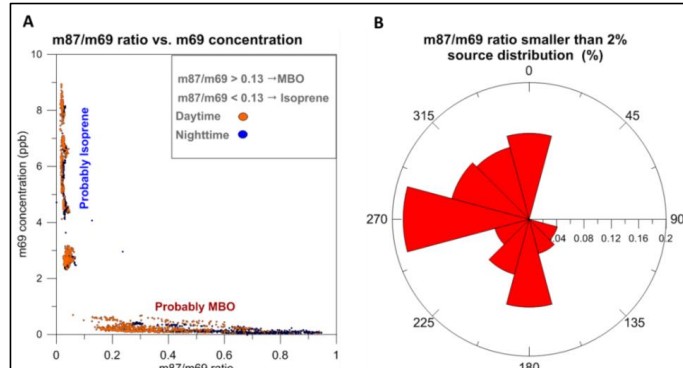

**Figure 6**. Isoprene and MBO origins. (a) Scatter plot of m69 mixing ratios as a function of the
m87/m69 ratio. Low and high ratios indicate a predominant contribution of MBO and isoprene,
respectively. The orange dots were measured during the daytime and the dark blue during the nighttime
(b) Fraction of time for each wind sector for which m87/m69 was <13%.

Two facts support isoprene+MBO predominant sea origin rather than the aquaculture
farms. First, back trajectories using HYSPLIT show no lower mixing ratios for
isoprene+MBO also in cases when the air masses were transported from the sea but
not over the aquaculture farms compared to transport of air masses over the



aquaculture (e.g., Fig. 5 and S11). Second, marine organisms have relatively short
life cycles, typically a few days (Tyrrell, 2001), and would likely have a variable
source strength from the aquaculture farms, which would not explain the similar
isoprene+MBO mixing ratios for different wind directions during a specific day. Our
measurements indicated no dependence of high isoprene+MBO mixing ratios on wind
direction during the day, reinforcing the sea's dominant role in isoprene emission,
rather than the aquaculture farms.
Interestingly, the isoprene mixing ratios during the nighttime remained
relatively high (~5-6 ppb) (Fig. 6a), the reason for which could be a relatively small
oxidative sink strength during the night. The daytime and nighttime isoprene lifetime
can be estimated based on its reaction with OH, $NO_3$ and $O_3$. We estimated the
average daytime OH and nighttime $NO_3$ concentrations, based on the MINOS
campaign in Finokalia, Crete (Berresheim et al., 2003; Vrekoussis et al., 2004), at
$4.5 \cdot 10^6 \frac{molec}{cm^3}$ (Berresheim et al., 2003), and $1.1 \cdot 10^8 \frac{molec}{cm^3}$ (Vrekoussis et al.,
2004), respectively. Using these concentrations, the reported rate constants for
isoprene with OH and $NO_3$ of $1 \cdot 10^{-10} \frac{cm^3}{molec \cdot sec}$ (Stevens et al., 1999) and $5.8 \cdot$
$10^{-13} \frac{cm^3}{molec \cdot sec}$ (Winer et al., 1984), respectively, and measured $O_3$ levels, we
obtained daytime and nighttime isoprene lifetimes of ~37 min and ~3.8 h,
respectively. This result points to isoprene emission occurring during the daytime.




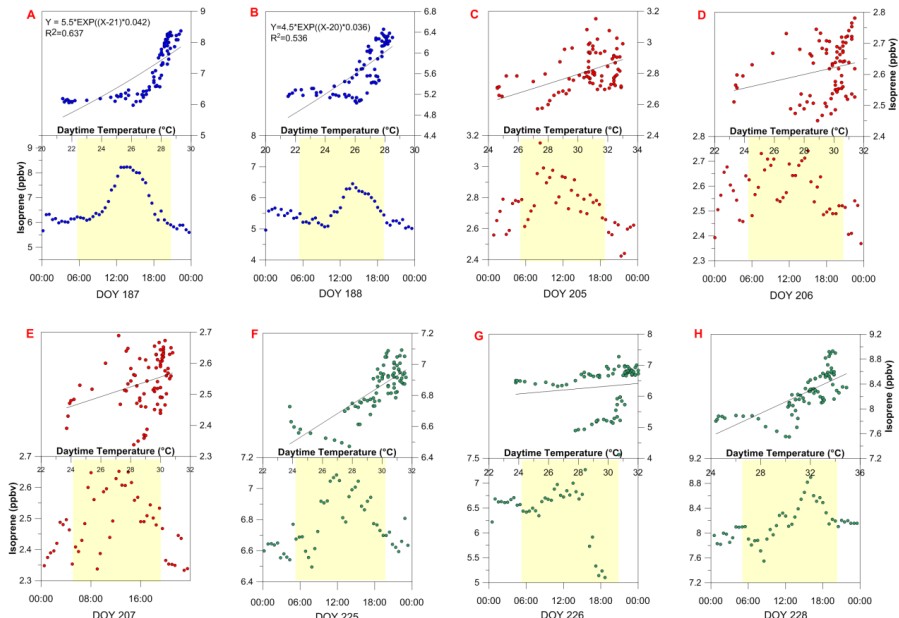


**Figure 7**. Isoprene (m87/m96<13%) mixing-ratio diurnal average and dependence on temperature.

Upper panels show regression between measured m87/m96<13% and T, and lower panels present that

of m87/m96<13%. Yellow shaded area represents daylight hours.

*3.2.2 Origin and characterization of DMS*

The discussion in Section 3.1 suggests that DMS is primarily emitted from the west,

pointing to a dominant marine emission source, with the less elevated mixing ratios

probably associated with emission from vegetation. According to the MEGANv2.1

simulation, the natural park's vegetation is a potent source of DMS (average

flux=0.477 $\frac{mg}{m^2 \cdot hr}$), slightly higher than the flux measured from insolated branches

(Jardine et al., 2015; Yonemura et al., 2005), while our analysis points to a stronger

emission from the memorial garden (see Fig. 4). As for isoprene, our analysis of DMS

mixing ratios with respect to wind direction rules out a significant contribution of the

aquafarms to the measured DMS, suggesting that the sea is a major source for DMS,





with apparently a strong dependency on T (Figs. 3, 4). DMS showed much less day-
to-day variations in its mixing ratios compared to isoprene and other BVOCs. This
corresponded with a clear day-to-day correlation of DMS mixing ratios with T. Figure
8 demonstrates a clear increase in the mixing ratios with T, throughout the
measurement period. Note that no significant dependency of DMS on global solar
radiation was observed.

The DMS mixing ratios peaked at ~0.18 ppbv. This figure is about an order of

magnitude lower than at the ocean surface (Tanimoto et al., 2014), about an order of
magnitude lower than in the Southern Ocean (Koga et al., 2014), slightly lower than
the maximum concentrations in the south Indian Ocean (Aumont et al., 2010), and
similar to the maximum concentrations on the coasts of Tasmania (Aumont et al.,
2010). Interestingly, the mixing ratios measured in this study are lower by about 1-2
orders of magnitude than those measured in the same region during August 1995
(Ganor et al., 2000), which could be attributed to a change in the marine biota as a
consequence of seawater warming.














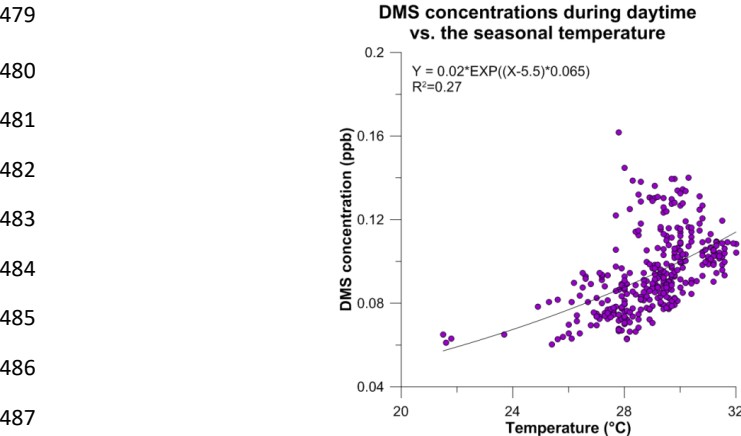

**Figure 8.** Daytime DMS mixing ratios from the western sector (marine source) as a function of the
temperature along the measurement campaign. An exponential fit between the two is included.

*3.2.3 Origin and characterization of* other BVOCs
Our findings in Figure 3 strongly suggest a common source for other BVOCs with
isoprene. We could not, however, use a wind-direction-based analysis to indicate
BVOCs' origin from the sea, since both sea and vegetation are located to the west of
the measurement point (see Fig. 1), and in contrast to isoprene, the other BVOCs were
indicated by MEGANv2.1 to be locally emitted. Furthermore, those BVOCs were less
variable with wind direction than was isoprene. We used MT summer measurements
from two other sites in Israel to assess whether MTs are likely to be transported to the
measurement site. According to MEGANv2.1, the average and maximal daytime MT
flux were $59 \frac{\mu g}{m^2 \cdot hr}$ and $152 \frac{\mu g}{m^2 \cdot hr}$ , respectively. While this predicted average flux is
lower than the mean MT measurements in the Birya and Yatir *Pinus halepensis Mill.*
forests in Israel (~200 $\frac{\mu g}{m^2 \cdot hr}$ and 800 $\frac{\mu g}{m^2 \cdot hr}$, respectively; Seco et al., 2017), the
corresponding measured mixing ratios in our study are generally higher than those
measured in those two sites, where in only a few cases the MT mixing ratios reached
above 0.5 ppbv for Birya, and the maximum was 0.2 ppbv in Yatir. Note that the





higher mixing ratios in our study, as compared with these two sites, were associated
with wind direction, either from the memorial garden or from the western sector (Fig.
4). This supports a relatively small local contribution of MTs in our study compared
with seawater.

3.3 *Concentrations of isoprene and DMS originating from the Levantine Basin*
Previous studies demonstrated the trade-off between DMS and isoprene in the marine
boundary layer, due to species distribution and climate, suggesting that most regions
are a source of either isoprene or DMS, but not both. While isoprene is emitted from
species that are more abundant in warmer regions and low-to-middle latitudes, DMS
is predominantly emitted in colder regions and higher latitudes (Dani and Loreto,
2017). This is in agreement with the relatively high isoprene/DMS mixing ratios in
our study. The SST in the Levantine Basin is relatively high, exceeding 30°C in
August 2015 at a distance of 2km from the coastline (IOLR, 2015). Further, SST
plays a significant role in determining which phytoplankton will dominate, and for a
given marine organisms population higher temperature and solar radiation tends to
enhance their BVOC emission, including DMS and isoprene (Dani and Loreto, 2017).
The strong emission of isoprene from the Levantine Basin can be attributed primarily
to its relatively high SST, considering the well-known correlation of isoprene
emission with SST (Dani and Loreto, 2017; Exton et al., 2013).

The relatively warm and oligotrophic sea enables cyanobacteria to become a

large fraction of marine primary production and phytoplankton (Krom et al., 2010;
Paerl and Otten, 2013; Pedrotti et al., 2017; Sarma, 2013) in the Levantine Basin, that
favors, in turn, emission of isoprene over other BVOCS, including DMS. Previous
measurements have indicated the presence of cyanobacteria in the Levantine Basin


during the summer of 2015 (Herut, 2016), with the cyanobacteria *Synechococcu*s and *Prochlorococcus* being the most abundant phytoplankton along the coasts of Israel during August 2015. A laboratory experiment demonstrated the emission of isoprene from the latter (Shaw et al., 2003). Other micro-organisms in the Levantine Basin (mostly dinoflagellates and diatoms) are generally less abundant. *Thalassiosira pseudonana* diatoms are also abundant along the coasts of Israel, which raises the possibility that the emission of isoprene from the sea is also influenced by this species. A laboratory experiment using PTR-MS indicated the emission of isoprene, as well as methanol, acetone and acetaldehyde from *Thalassiosira pseudonana* diatoms, but isoprene is the only one among these that is not consumed by bacterioplankton within the water column (Halsey et al., 2017).

DMS can be also emitted by diatoms, but at lower rates under warmer conditions (Dani and Loreto, 2017; Levasseur et al., 1994). In addition, DMS is a common microbial VOC, formed in various marine environments by bacterial decomposition of dimethylsulfoniopropionate (DMSP) (Bourne et al., 2013; Howard et al., 2008). DMS in the marine boundary layer is mostly emitted by dinoflagellates and haptophyte coccolithophores. Dinoflagellates, as well as *Thalassiosira pseudonana* diatoms, were constantly observed along the coast in estuary zones several kilometers from the measurement site (Herut, 2016). This might explain the relatively minor day-to-day variations in the mixing ratios of DMS (Fig. 3), which, in turn, resulted in a relatively high correlation of the mixing ratios with T throughout the measurement periods. Cyanobacteria blooms and collapses depend on the nutrient supply and have no seasonality (Paerl and Otten, 2013), which can be an additional reason for the fluctuations in isoprene.



### 4. Conclusions


Our findings indicate that high isoprene emission from the Eastern
Mediterranean Sea contributes up to ~9 ppb several km inland from the sea shore. The
apparently strong emission of isoprene can be attributed primarily to the relatively
high SST of the Levantine Basin, considering the well-known correlation of isoprene
emission with SST growth conditions (Dani and Loreto, 2017; Exton et al., 2013).
Furthermore, isoprene mixing ratios tended to strongly increase with diurnal increases
in T, but there was no correlation with solar radiation. Our analysis points to
cyanobacteria as a dominant source for the isoprene emission, as are other possible
marine microbiomes, supporting previous findings (Arnold et al., 2009; Bonsang et
al., 2010; Dani and Loreto, 2017; Hackenberg et al., 2017; Shaw et al., 2003).
Measured DMS mixing ratios were lower by 1-2 orders of magnitude than those
measured in 1995 (Ganor et al., 2000) in the same area during the same season,
suggesting a strong impact of SST on the decadal change in DMS emissions via
changes in species composition. Considering that, according to IPCC, ocean SST is
expected to rise by 5°C by the year 2100 (Hoegh-Guldberg et al., 2014), efforts are
required to adequately represent the complex dependency of marine BVOC emissions,
such as isoprene and DMS, on SST, to improve the predictability of both air-quality
and climate models. Our study results indicate that this increase in SST can
significantly increase the emission of isoprene into the MBL. This can greatly affect
air quality, considering its high photochemical reactivity, with particularly negative
implications in urbanized coastal areas, where on-shore wind typically occurs during
the daytime, controlled by the sea-land breeze. Furthermore, elevated isoprene
emission is expected from coastal areas where coastal upwelling can significantly



affect biological activity, which was shown to correlate with BVOC emissions (Gantt
et al., 2010).

Comprehensive evaluation of the impact of marine organism emissions on

both the atmospheric chemistry and radiative budget should rely on a suite of gases.
Along with the high isoprene levels, relatively low DMS mixing ratios were observed
under the studied conditions, which supports previous studies that have indicated a
general contrasting spatial distribution, partially controlled by SST and latitude
(Yokouchi et al., 1999) and lower DMS emission under relatively low temperature
(Dani and Loreto, 2017). While DMS and isoprene emissions are influenced in a
contrasting manner by changes in SST, both tend to rise in response to a SST increase
for a given phytoplankton population (Dani and Loreto, 2017), as supported by this
study.

Significant contribution of oceanic emission of other BVOCs, such as acetone,

acetaldehyde and monoterpenes have been also reported by previous studies. We
found supporting indications for dominant emission of MT from the Levantine Basin,
further suggesting significant emission of other BVOCs from this source. The
analyses also indicate that estuaries play a potentially important role in facilitating the
emission of DMS, and probably additional BVOCs, by maintaining a suitable
environment for phytoplankton growth. In agreement with a previous study (Goldstein
et al., 2004), our analyses suggest that thinning may play an important role in
facilitating BVOC emissions, a mechanism which should be taken into consideration
especially in urban areas with cultivated parks and gardens.

This study demonstrates that most of the VOCs studied here are controlled by

both anthropogenic and marine and terrestrial biogenic emission sources, highlighting
the need for the strict identification of the origin and representative models for both



emission source types. Our study further highlights the Levantine Basin's capability
to serve as a natural laboratory for studying both anthropogenic stress and climate
change on marine BVOC emissions. More comprehensive research is required to
directly address the impact of oligotrophication and increased SST on marine BVOC
emissions.

**Data availability**. Data are available upon request from the corresponding authors
Eran Tas (eran.tas@mail.huji.ac.il) and Erick Fredj (erick.fredj@gmail.com).

**Author contribution**. ET designed the experiments, MG and GL carried the field
measurements out and PM and EF led the calibration, quality control and data
processing. AG setup the MEGANv2.1 model. CD and ET led the analyses with
contributions from all co-authors. ET and CD prepared the manuscript with
contributions from all co-authors.

**Competing interests**. The authors declare that they have no conflict of interest**.**

**Acknowledgements**
We want to greatly thank the crew of Ramat Hanadiv and Gil Lerner for supporting
the measurements. This study was supported by the Israel Science Foundation, Grant
No. 1787/15. E.T. holds the Joseph H. and Belle R. Braun Senior Lectureship in
Agriculture.



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
