# Peer review of "Emission of biogenic volatile organic compounds from warm and oligotrophic seawater at the Eastern Mediterranean"

_Atmospheric Chemistry and Physics, 2019_

## Referee Comment (RC1) · Silvano Fares (Referee) · 6 Feb 2020

The authors tried to determine sources of BVOCs emitted from a mixed vegetation site 4 km away from the coast of Levantine Basin. They demonstrate relevant biogenic sources frm inland but also from the ocean, and explained different sources of BVOC species with the help of transport models and emission models. I believe this paper helps understanding the complex synamics of Biogenic emission and transport. There are no language flaws and a good amount of references. Showing measured fluxes would have been beneficial to better understand didirectianal BVOC fluxes at the measuring site. A list of minor comments is reported here:

[Figure]

INTRO Lines 113-115: can you explain why an oligotrophic environment is represented by unicellular organisms and plankton?

M&M Line 173-195: Although the authors cite other previous papers, since the manuscript is all about detected BVOCs, it is important to provide more details on the calibration procedure. Line 208: why are you recording slow sensors at 10Hz?

Results Lines 271-279: Althogh grouping results and discussion may not be ideal, please insert at least numbers (with SD) while discussing mixing ratios. What does extreme mixing ratios mean?

While reading these interesting seasonal variations of BVOCs the reader wonders why you did not show fluxes considering that you have an Eddy Covariance installation at the site and the PTR-TOF-MS allows very fast measurements. Fluxes, when available, may support understanding of BVOC origin, way better than modelling.

Lines 379-281: although I understood the sense of this sentence, it may be written more clearly to stress that high mixing ratio corresponding to Isoprene + MBO is a proxy of high isoprene emission. Line 446: a lifetime of 3.8 hrs does not really support the possibility that isoprene is only emitted during the day. Considering the light dependency of isoprene emission this may be consumed early in the night.

Line 499: since you are comparing MEGAN results with previous analysis, you should enter more into detail on what species drive emission, assuming that some of the species present at your measuring site and more inland are described and characterized in some papers for their emission capacity. Perhaps MEGAN adopts a wide plant functional type?

———————————————

---

## Referee Comment (RC2) · Anonymous Referee #2 · 23 Mar 2020

The authors report BVOC measurements at a site close to the Mediterranean sea. They suggest that a large amount of the BVOCs originate from the sea. This is a result that merrits publication. However, the evidence presented is not very consistent and/or robust in several aspects (see specific comments). The authors use (maybe too) many tools (PMF, Hysplitt, MEGAN, WRF) to substantiate their case, however, for most of these methods insufficient information is given to judge their appropriate application. The paper needs a substantial revision before publication. I recommend to focus on a careful and clear presentation of the strongest evidence. I wonder why the authors did not use the eddy covariance technique to constrain local emissions.

[Figure]

Specific comments:

line 78: "... are ESTIMATED to be substantially smaller ..." I guess nobody really knows the marine source strength.

Fig 1: remove the red triangle that separates panel a and b. This is confusing.

Table S1: what is meant by poor quality and failed calibration?

There is no Figure 2.

Caption of Fig S4 can be improved. What is the color code?

Why are benzene, toluene, and acetonitrile not reported? These compounds should be valuable tracers to constrain traffic emissions (2 highways between the site and the sea!) and biomass burning.

The evidence shown in table 1 is insufficient to classify the period September 14-17 as irregular conditions. PAR is less than 20% lower than in the other period in September. All other parameters are similar.

Lines 277-280: I don't see extreme meteorological conditions and I don't see extreme concentrations in Fig 3. The modelling exercise in the supplement is not convincing because there is no reference period. It could be convincing, for example, if much higher boundary layers would be calculated for the second period in September.

Lines 281-283: Please explain why the shape suggests a biogenic source. Also, it would be useful to see the data for DMS and other BVOCs to show their difference!

Line 303: would be good to see hexenal and hexanal in Fig 4, even if these compounds are not calibrated!

309-312: the fact that MEGAN does not predict local isoprene emissions is no convincing argument. Surely not all species (including invasive species) are included in the MEGAN model.

315: I cannot follow all details of the kinetic analysis in the supplement, but I doubt that this can rule out the possibility of local emissions. I do not understand why the authors do not process their data with the eddy covariance technique. This would give a clear answer on whether there are local emissions or not.

324: I do not agree that this has been demonstrated...

327:Figure S12: it would be informative to see how the scatter plot looks like for other VOCs.

328: Figure S10 shows data for one day. This is insufficient to make general claims.

334: Fig 5 caption: There is no "regression" you just show a scatter plot temp vs iso+MBO

369-370: Given the fact that you measured at high E/N (140 Td) I am not so sure that such high fractions of MBO are expected at 87 Th. Maybe you can prove this by showing calibration measurements.

445-447: I think that it would be interesting to estimate daytime isoprene production from these lifetime values.

475-478: what was the sea surface temperature in 1995 as compared to 2015?

---

## Author Response (AR1)

Dear Editor,

We are pleased to submit a revised version of the manuscript (acp-2019-1170) "Emission of biogenic volatile organic compounds from warm and oligotrophic seawater at the Eastern Mediterranean" for publication in Atmospheric Chemistry and Physics.

We thank the referee Dr. Silvano Fares and the anonymous reviewer for investing time and effort in reading our manuscript, and for the useful comments which helped us to greatly improve its quality, both scientifically and for clarity. We realized that the various analyses presented in the original version to indicate strong emission of isoprene from the Eastern Mediterranean Sea were not clear or sufficiently complete. Following suggestions by reviewer II on this issue, we reduced the number of tools used to support the strong isoprene emission from the seawater, and demonstrated that our arguments are valid for all measurement periods. The latter was done mainly by adding analysis results to the Supplement to support our arguments for all measurement periods, in an attempt to keep the main text succinct. In addition, we divided the section on the origin apportioning of measured isoprene (section 3.2.1) into several subsections to clarify the logic behind the related analyses.

We seriously considered and addressed all of the reviewers' comments. In the following, we present detailed point-by-point responses to these comments and detail the changes implemented in the revised version. These are followed by the revised marked-up manuscript. We hope that the revised version will be found suitable for publication in Atmospheric Chemistry and Physics.

Sincerely,

       Eran Tas

**Responses to comments by Dr. Silvano Fares**

We deeply thank the reviewer for the effort invested in reviewing this paper and for its thorough and constructive review. In the following, we present detailed point-by-point responses to the comments by Dr. Fares.

*The authors tried to determine sources of BVOCs emitted from a mixed vegetation site 4 km away from the coast of Levantine Basin. They demonstrate relevant biogenic sources frm inland but also from the ocean, and explained different sources of BVOC species with the help of transport models and emission models. I believe this paper helps understanding the complex synamics of Biogenic emission and transport. There are no language flaws and a good amount of references. Showing measured fluxes would have been beneficial to better understand didirectianal BVOC fluxes at the measuring site. A list of minor comments is reported here:*

*INTRO Lines 113-115: can you explain why an oligotrophic environment is represented by unicellular organisms and plankton?*

**Answer:**

The oligotrophic environment is characterized by limited nutrients. Small cells are basically more effective at nutrient uptake (Fogg, 1986 ), while unicellular organisms are more efficient at $CO_2$ fixation (Mazard et al., 2004). This gives rise to both the unicellular and small plankton being better adapted to oligotrophic conditions. In addition, high seawater temperature tends to shift the planktonic community toward an increase in unicellular and small plankton (Mazard et al., 2004;Rasconi et al., 2015).

We provide these explanations in the revised manuscript and have revised the corresponding paragraph as follows: "The Eastern Mediterranean Basin region has been recognized as highly responsive to climate change, and has been aptly named a primary "climate change hotspot" (Giorgi, 2006; IPCC, 2007; Lelieveld et al., 2012). This makes it an attractive site to study the impact of anthropogenic stress and climate

change on marine BVOC emissions. In addition, being oligotrophic, there is a predominance of unicellular and small plankton such as cyanobacteria (Krom et al., 2010) that can more efficiently perform $CO_2$ fixation and utilize nutrients under such conditions, respectively(Fogg, 1986 ;Mazard et al., 2004). Moreover, the high SST tends to further shift the planktonic community toward an increase in unicellular and small plankton (Mazard et al., 2004;Rasconi et al., 2015)."  (lines 111-120).

*M&M Line 173-195: Although the authors cite other previous papers, since the manuscript is all about detected BVOCs, it is important to provide more details on the calibration procedure.*

**Answer:**

We have added more information on the calibration procedure in Sect. 2.2: " The PTR-ToF-MS was calibrated every 1-2 days for background (zero), and weekly for sensitivity (span), subject to technical limitations (see Table S1). Background (zero) calibration was conducted by sampling ambient air which was passed through a catalytic converter heated to 350°C. Sensitivity calibration was performed using gas standards (Ionicon Analytik GmbH, Austria) containing methanol (0.99±8% ppmv), acetonitrile (0.99±6% ppmv), acetaldehyde (0.95±5% (ppmv), ethanol (1.00±5% ppmv), acrolein (1.01±5% ppmv), acetone (0.98±5% ppmv), isoprene (0.95±5% ppmv), crotonaldehyde (1.01±5% ppmv), 2-butanone (0.99±5% ppmv), benzene (0.99±5% ppmv), toluene (0.99±5% ppmv), o-xylene (1.02±6% ppmv), chlorobenzene (1.01±5% ppmv), α-pinene (1.01±5% ppmv) and 1,2-dichlorobenzene (1.02±5% ppmv) to obtain gas mixtures ranging from 1-10 ppbv. Mixing ratios of compounds for which no gas standard was available were calculated using default reaction rate constants (see Sect. S1)." (lines 191-203).

*Line 208: why are you recording slow sensors at 10 Hz?*

**Answer:**

This was a typo. We have revised the text as follows: "These measured data were recorded with a CR10X data logger (Campbell Scientific) at 10-min frequency" (lines 227-229).

*Results Lines 271-279: Althogh grouping results and discussion may not be ideal, please insert at least numbers (with SD) while discussing mixing ratios.*

**Answer:**

Done. The text which appeared originally on lines 271-279 has been revised as follows: " DMS showed the strongest correlation with the average daytime temperature ($r^2$=0.27; see Sect. 3.2.2), corresponding to a significant increase in the mixing ratios between early summer (0.072±0.005 ppb, day of year (DOY) 188) and the end of summer (0.19±0.040 ppb, DOY 254), which decreased during the autumn (0.17±0.015 ppb, DOY 255 to 0.066±0.011 ppb, DOY 283). The other BVOCs, except for isoprene+MBO, showed a gradual increase in their average mixing ratios during the summer and early autumn (DOY 198-269; acetone from 3.74±0.767 ppbv to 4.33±0.471 ppbv, acetaldehyde from 1.64±0.595 ppbv to 3.09±0.496 ppbv, MT from 0.089±0.021 ppbv to 0.237±0.120 ppbv, MVK+MACR from 0.125±0.048 ppbv to 0.252±0.070 ppbv), and lower average mixing ratios in the autumn and early winter (DOY 270-286; DMS 0.091±0.026 ppbv, acetone 3.96±1.04 ppbv, acetaldehyde 1.86±0.97 ppbv, MT 0.139±0.064 ppbv, isoprene+MBO 0.182±0.093 ppbv, MVK+MACR 0.153±0.098 ppbv), which can be explained by the correlation with air temperature (Fig. 2). During DOY 257-260, BVOCs showed elevated mixing ratios (daytime averages for DMS, acetone, acetaldehyde, $H_2S$, MT, isoprene+MBO and MVK+MACR were 0.122±0.016 ppbv, 13.6±3.26 ppbv, 8,138±1.18 ppbv, 0.046±0.021 ppbv, 1.97±0.215 ppbv, 7.68±0.218 ppbv and 0.644±0.084 ppbv, respectively), as well as irregular diurnal shape, which may be attributed to synoptic-scale induced processes (see Sect. S6). We therefore did not use these measurements for further analyses." (lines 290-308).

*What does extreme mixing ratios mean? While reading these interesting seasonal variations of BVOCs the reader wonders why you did not show fluxes considering that you have an Eddy Covariance installation at the site and the PTR-TOF-MS allows very fast measurements. Fluxes, when available, may support understanding of BVOC origin, way better than modelling.*

**Answer:**

We have changed "extreme mixing ratios" to "elevated mixing ratios […] as well as irregular diurnal shape, which may be attributed to synoptic-scale induced processes (see Sect. S6)." (lines 303-307). The irregular diurnal shape may be attributed to synoptic-scale induced subsidence and intrusion (see Sect. S6), and therefore, we did not use these measurements for further analyses.

Unfortunately, we encountered technical problems in evaluating flux for the measured VOCs using the rented PTR-ToF-MS (mostly systematic errors in writing the data in 10 Hz frequency, negating the possibility of making cross-correlations). We agree that mentioning eddy covariance measurements without showing BVOC fluxes can be confusing. We therefore address this point in the revised version. To provide a full and proper description of the measurement setup, we retain the information on the eddy covariance in the revised version (used, for instance, to evaluate $CO_2$ flux; Sect. S3 in the Supplement). Overall, the revised text in Sect. 2.2 reads as follows: "The set of instruments included a platform for eddy covariance measurements of BVOCs, $O_3$, carbon dioxide ($CO_2$) and water vapor ($H_2O$), trace-gas mixing ratios, including $O_3$, $NO_X$, $SO_2$ and CO, and basic meteorological conditions, using an air-conditioned mobile laboratory and two towers (Fig. S2). Note that due to technical problems, VOC fluxes were not evaluated." (lines 171-175). We also mention the lack of VOC flux evaluation in the caption of Fig. S2 in the Supplement.

*Lines 379-281: although I understood the sense of this sentence, it may be written more clearly to stress that high mixing ratio corresponding to Isoprene + MBO is a proxy of high isoprene emission.*

**Answer:**

We have removed this section, based on reviewer II's recommendation to focus on the strongest evidence for our analyses.

*Line 446: a lifetime of 3.8 hrs does not really support the possibility that isoprene is only emitted during the day. Considering the light dependency of isoprene emission this may be consumed early in the night.*

**Answer:**

We agree and have made it clear that a lifetime of 3.8 h does not rule out isoprene nighttime emission: "Considering the relatively moderate decrease in the measured isoprene during the night (Figs. S12-S17 in the Supplement), this result points to stronger isoprene emissions during daytime, but does not rule out nighttime isoprene emissions" (lines 472-475). Note that pursuant to a comment by reviewer II, we support relatively weak nighttime isoprene production with a simplified kinetic calculation: "A rough estimation of isoprene production rate can be calculated by subtracting the isoprene loss rate, evaluated from its calculated lifetime, from its measured mixing ratios. These simplified calculations indicate a daytime and nighttime isoprene production rate ranging between $\sim 4.9 \cdot 10^{-5}$ and $1.7 \cdot 10^{-2}$ ppbv $\cdot$ $s^{-1}$ (average $5.2 \cdot 10^{-3} \pm 5.6 \cdot 10^{-3}$ ppbv $\cdot s^{-1}$) and between $-1.3 \cdot 10^{-3}$ and $1.3 \cdot 10^{-3}$ (average $-1.6 \cdot 10^{-6}$ ppbv $\cdot s^{-1}$ $\pm 1.4 \cdot 10^{-5}$ ppbv $\cdot s^{-1}$), supporting a much smaller isoprene production rate during the night vs. daytime" (lines 476-482).

*Line 499: since you are comparing MEGAN results with previous analysis, you should enter more into detail on what species drive emission, assuming that some of the species present at your measuring site and more inland are described and characterized in some papers for their emission capacity. Perhaps MEGAN adopts a wide plant functional type?*

**Answer**:

MEGAN emissions were driven by emission factors for four specific vegetation species (*Quercus calliprinos* (25%), *Pistacia lentiscus* (20%), *Rhamnus lycioides* (2%), *Pinus halepensis* (<5%); see species composition description on lines 149-155). For two species (*Phillyrea latifolia* (7.5%) and *Cupressus* sp. (5%)), emissions are based on genus level data which are expected to be fairly representative, while for another two (*Sarcopoterium spinosum* (~2%) and *Calicotome villosa* (1%)), the emission factors were calculated based on average emission at the family level (Rosaceae and Fabaceae, respectively), which may lead to larger inaccuracies.

We evaluated the contribution of different species to monoterpene emissions at Ramat Hanadiv and updated the discussion with this information, which compares the ratio between monoterpene emission flux and its mixing ratios with two other *Pinus halepensis* forests: "We used the ratio between MT flux and mixing ratio at the three sites as a basis to address this inquiry. Note that according to the MEGAN v2.1 simulations (see Sect. 2.3), the MT emissions in Ramat Hanadiv were driven by *Quercus calliprinos* (48.1%), *Pistacia lentiscus* (19.8%), *Phillyrea latifolia* (7.12%) and *Cupressus* spp. (6.17%), as well as other species (see Sect. S5), in contrast to the two *Pinus halepensis* plantations, Birya and Yatir. While the fact that MT is not emitted by the same vegetation species should not significantly affect our analysis, we recognize that there may be differences in the MT composition and atmospheric oxidation capacity at the three sites which would influence MT lifetimes and lead to some differences in the flux-to-concentration ratios." (lines 540-549).

**References:**

Fry, J. C.: Oligotrophs. In Microbiology of extreme environments, edited by C. Edwards, Open University Press., 1990.

Platt, T., Rao, D. V. S. and Irwin, B.: Photosynthesis of picoplankton in the oligotrophic ocean, Nature, 301(5902), 702–704, doi:10.1038/301702a0, 1983.

Fogg, G. E.: Picoplankton, Proc R Soc Lond,, B, :l-30 1986

Mazard, S. L., Fuller, N. J., Orcutt, K. M., Bridle, O., and Scanlan, D. J.: PCR analysis of the distribution of unicellular cyanobacterial diazotrophs in the Arabian Sea, Appl Environ Microb, 70, 7355-7364, 10.1128/Aem.70.12.7355-7364.2004, 2004.

Rasconi, S., Gall, A., Winter, K., and Kainz, M. J.: Increasing Water Temperature Triggers Dominance of Small Freshwater Plankton, Plos One, 10, ARTN e014044910.1371/journal.pone.0140449, 2015.

**Response to comments by reviewer #2**

We deeply thank the reviewer for the effort invested in reviewing this paper and for its thorough and constructive review. In the following, we present detailed point-by-point responses to the comments by the reviewer.

*The authors report BVOC measurements at a site close to the Mediterranean sea. They suggest that a large amount of the BVOCs originate from the sea. This is a result that merits publication. However, the evidence presented is not very consistent and/or robust in several aspects (see specific comments). The authors use (maybe too) many tools (PMF, Hysplitt, MEGAN, WRF) to substantiate their case, however, for most of these methods insufficient information is given to judge their appropriate application.*

*The paper needs a substantial revision before publication. I recommend to focus on a careful and clear presentation of the strongest evidence.*

**Answer:**

We realize that the analyses used to support isoprene and DMS from the seawater were not presented clearly enough, and we have therefore revised Sects. 3.2.1 and 3.2.2 to clarify our arguments. In particular, the analyses presented in Sect. 3.2.1 are complex and use several tools. In the absence of available measured VOC flux (see our answer to the next comment), we had to use several independent methods to provide strong supporting evidence for the origin of isoprene from the Mediterranean Sea.

In response to this comment, we excluded the analysis by PMF as well as that of MEGAN v2.1 to support the partitioning of "isoprene+MBO". In addition, to clarify our analysis-based arguments, we divided section 3.2.1 into several

subsections. The new subsections and various analyses are integrated in Sect. 3.2.1 as follows: i) "*Potential anthropogenic emission sources of isoprene+MBO*" – ruling out any significant anthropogenic contribution from traffic based on significant differences in diurnal profile for isoprene and MT vs. the diurnal profiles of benzene, toluene, acetonitrile (Figs. S12-S17 in the Supplement) and isoprene vs. CO (Fig. S19 in the Supplement for DOY 225); ii) "*Potential biogenic emission sources of isoprene+MBO*" – supporting evidence for dominant emission from a biogenic source based on correlation with temperature, but highlighting the need to partition the isoprene+MBO signal, due to an insufficiently clear association between this signal (i.e., m/z=69) and the measured air temperature; iii) "*Partitioning of isoprene+MBO signal*" – partitioning of the "isoprene+MBO" signal based on m87/m67 fractionation; iv) "*Isoprene origin*" – supporting a dominant biogenic source for the partitioned isoprene (m87/m69 < 2%) based on its correlation with temperature (Fig. 6 in the main text and Figs. S12-S17 in the Supplement), indicating a marine source for isoprene based on analysis of wind direction vs. m87/m69 < 2% (as well as the high day-to-day variation in isoprene mixing ratios as depicted in Fig. 6 in the main text), ruling out a significant contribution of aquaculture farms to the detected isoprene based on HYSPLIT back trajectories (Fig. S20 in the Supplement), and robustness of isoprene mixing ratios to changes in wind direction during each measurement day.

*I wonder why the authors did not use the eddy covariance technique to constrain local emissions.*

**Answer:**

We applied the eddy covariance technique to evaluate the flux of $CO_2$, $H_2O$ and $O_3$ ($CO_2$ flux was used to support drought effects on BVOC emission; Fig. S18 in the Supplement). Unfortunately, we could not evaluate VOC fluxes due to technical problems with the high-frequency (10 Hz) recording of the measured data, which was inconsistent. We realize that a statement to this effect was missing in the text, and we have therefore added the following: "Note that due to technical problems, VOC fluxes were not evaluated" (line 175). We also mention the absence of VOC flux evaluation in the caption of Fig. S2 in the Supplement.

*Specific comments:*

*line 78: "... are ESTIMATED to be substantially smaller ..." I guess nobody really knows the marine source strength.*

**Answer:**

We agree, and have changed the sentence accordingly: "Although the emission rates of isoprene into the marine boundary layer (MBL) are estimated to be substantially smaller than terrestrial emissions…" (lines 77-79).

*Fig 1: remove the red triangle that separates panel a and b. This is confusing.*

**Answer:**

Done

*Table S1: what is meant by poor quality and failed calibration?*

**Answer:**

"Failed calibration" is defined in the revised Supplement as follows: "...data which could not be properly calibrated due to failed calibration" (Supplement lines 32-33). In most cases, failed calibration occurred due to accidental dilution of the calibrated mixture with ambient air. "Poor quality" is defined in the revised manuscript as follows: "…data that corresponded with unrealistic mixing ratios or erroneous recording..." (Supplement lines 34-35). The latter refers to data recording that repeatedly resulted in the same value.

*There is no Figure 2.*

**Answer:**

Fixed

*Caption of Fig S4 can be improved.*

**Answer:**

The figure caption has been amended (lines 274-277 in the Supplement).

*What is the color code?*

**Answer:**

We have added a legend to the figure, as well as for Figs. 4 and 6 in the main text. Each color indicates a specific period of 2-4 sequential days (as summarized in Table 1).

*Why are benzene, toluene, and acetonitrile not reported? These compounds should be valuable tracers to constrain traffic emissions (2 highways between the site and the sea!) and biomass burning.*

**Answer:**

In Sect. S3, we have added figures that compare the diurnal profiles of MT and isoprene+MBO with those of benzene, toluene and acetonitrile. In Sect. 3.2.1, we refer the reader to those figures (lines 366-370). Note that as discussed in Sect. S3 (lines 191-192), the correlation of acetonitrile with acetone and acetaldehyde suggests that the former is also emitted from a biogenic source. This was most salient for 13-14 August (see Fig. 1 below).

[Figure]

**Figure 1**. *Average diurnal profiles for acetone, acetonitrile, acetaldehyde and $CO_2$ fluxes during 13-14 August 2015 (DOY 225-226). Filled circles and vertical bars represent the mean and standard deviation of the mixing ratios, respectively. The anticorrelation between $CO_2$ and relatively soluble BVOCs suggests limitation of the BVOCs due to the drought effect (see Fig. 4 below and related discussion). The figure suggests a biogenic contribution for acetonitrile, based on its association with acetaldehyde and acetone, and apparent response to midday drought impact.*

*The evidence shown in table 1 is insufficient to classify the period September 14-17 as irregular conditions. PAR is less than 20% lower than in the other period in September. All other parameters are similar.*

**Answer:**

We agree and address this point, following additional analyses, as described in the answers to the next two comments.

*Lines 277-280: I don't see extreme meteorological conditions and I don't see extreme concentrations in Fig 3.*

**Answer:**

We have revised this according to our response to the next comment as follows: " During DOY 257-260, BVOCs showed elevated mixing ratios […], as well as irregular diurnal shape, which may be attributed to synoptic-scale induced processes (see Sect. S6). We therefore did not use these measurements for further analyses." (lines 302-308).

*The modelling exercise in the supplement is not convincing because there is no reference period. It could be convincing, for example, if much higher boundary layers would be calculated for the second period in September.*

**Answer:**

In comparing the boundary layer height (BLH) with the corresponding BLH for the second period in September, as well as for the whole month, we realized that BLH for DOY 257-260 was not significantly smaller than average, except for DOY 257 for which the radiation was lower than for DOY 258-260 (see Fig. S22 in the Supplement).

In addition to the elevated mixing ratios during DOY 257-260, the diurnal profiles of the BVOCs for this period differed from other periods by not showing any clear increase during the day, and starting to continuously decrease from about 0800–0900 h, except for DMS, where the decrease started earlier (see Fig. 3 below and Fig. S7 in the Supplement). Our new analysis indicates subsidence of air from the upper troposphere which is typical for the studied area during the summer, leading to a significant change in the mixing ratios (e.g., of $O_3$) within the boundary layer, via air exchange with the upper troposphere along with the subsidence (e.g., (Tyrlis and Lelieveld, 2013;Zanis et al., 2014;Li et al., 2018). Such subsidence can frequently lead to shallowing of the BLH. As evidenced by our model simulations, DOY 257-260 were characterized by notably early (~0900 h) shallowing of the BLH (see Fig. 2 below), strongly supporting the occurrence of subsidence for DOY 257-260.

The fact that anthropogenic trace gases also showed a similar diurnal profile, but with an earlier decrease in mixing ratios during the morning (Fig. S7 in the Supplement) suggests that the anthropogenic trace gases were also significantly diluted by the subsidence which was apparently accompanied by intrusion. We attribute the delayed and more moderate decrease for BVOCs compared to anthropogenic VOCs to an increase in emission in response to increasing temperature and radiation intensity for the former. Our analysis is insufficient to conclusively explain the elevated BVOC mixing ratios and irregular diurnal profile during DOY 257-260. Nevertheless, we exclude this period from the rest of the analyses (except in Fig. 2 in the main text) as in the original version of the manuscript, due to the irregular diurnal profile. We have revised our explanations of the elevated mixing ratios and irregular diurnal shape for this period, according to the above (see Sect. S6).

[Figure]

**Fig. 2**. Average diurnal profile of the boundary layer height (BLH) during DOY  268-269 (upper panel) and DOY 257-260 (lower panel). Gray vertical error bars represent standard deviation.

[Figure]

**Fig. 3**. *Daily average diurnal profile for selected VOCs which were dominated by biogenic sources (monoterpenes (MT), isoprene+2-methyl-3-buten-2-ol (MBO), dimethyl sulfide (DMS), acetone, acetaldehyde and the sum of methyl vinyl ketone and methacrolein (MVK+MACR)) and anthropogenic sources (1,3-butadiene, H$_2$S, NO$_X$ and SO$_2$) for 14-17 September 2015. Filled circles and vertical bars represent mean mixing ratios and their standard deviation, respectively.*

*Lines 281-283: Please explain why the shape suggests a biogenic source. Also, it would be useful to see the data for DMS and other BVOCs to show their difference!*

**Answer:**

This is now explained in the text as follows: "The diurnal profile of isoprene+MBO suggests a predominantly biogenic source due to a clear daytime increase and a correlation with temperature for most of the periods (Fig. 4, Figs. S3-S9). However, its day-to-day mixing ratios showed higher variability (Fig. 2), which was quite different from both DMS and the other BVOCs. The origin of the BVOCs is explored in the next section" (lines 309-313). Note that isoprene+MBO differed from DMS and other BVOCs in the day-to-day variation, as is evident in Fig. 2 (above), and supported by calculations (see Table 1 below).

Note also that acetaldehyde and acetone, and to a lesser extent MVK+MACR, showed somewhat less typical biogenic diurnal shapes compared to the other BVOCs

during some of the measurement periods. This can be explained by the former species' higher solubility compared to the other BVOCs that we investigated, making them significantly more susceptible to drought effects via stomatal activity, as compared to non-soluble BVOCs (Niinemets et al., 2004;Niinemets et al., 2014). This is demonstrated, for instance, for 13-14 August (see Fig. S11 in the Supplement). The figure indicates an increase in the downward $CO_2$ flux after sunrise, until ~0930 h, then a moderate decrease in this flux until ~1500 h which can be attributed to a drought-induced midday depression in the photosynthetic rate. This apparent midday depression was followed by an additional peak in the downward $CO_2$ flux at around 1600 h. Acetaldehyde and acetone showed a decrease in mixing ratios between ~0900 and 1600 h along with the decrease in the downward flux of $CO_2$, and peaked again at ~1600 h, suggesting a strong limitation of their emission due to the drought conditions. MVK+MACR did not show a peak at ~1600 h, and it was less clearly affected by the midday depression compared to acetaldehyde and acetone.

[Figure]

**Fig. 4**. *Average diurnal profiles for acetone, acetonitrile, acetaldehyde and $CO_2$ flux during 13-14 August 2015 (DOY 225-226). Filled circles and vertical bars represent the mean and standard deviation of the mixing ratios, respectively. Yellow shaded area represents daytime. The anticorrelation between $CO_2$ and soluble BVOCs, during the daytime, suggests limitation of the BVOCs emission due to the drought effect. In particular, elevated mixing ratios of acetaldehyde and acetone between ~0700 and 0900 h, and between ~1600 and 1700 h, and lower mixing ratios in between, are in line with the trends of $CO_2$, suggesting a midday depression in photosynthesis as well as in the emission of the four relatively soluble VOCs in response to a drought effect (see (Niinemets et al., 2014;Niinemets et al., 2004)).*

*Line 303: would be good to see hexenal and hexanal in Fig 4, even if these compounds are not calibrated!*

Hexenal and hexanal can provide an indication of green leaf volatile (GLV) emissions following wounding. Considering that this aspect is not central to our study, we included it in the Supplement (see Sect. S7). A source analysis similar to that

presented in the original Fig. 3 was also performed for hexanal (m/z=83.085), hexenal (m/z=57.033 and m/z=99.080) and methanol (m/z=33.034), and the reader is referred to this analysis from the main text (lines 328-331). Note that hexenal is also detected at m/z=43.018 and m/z=81.070 (Brilli et al., 2011; Pang, 2015), but we did not include these fragments in the hexenal mixing ratio calculations because m/z=43.018 can be affected by 1-hexyl acetate and other GLV fragments (m/z=43.018), while m/z=81.080 can be affected by MTs. Hexanal is also detected at m/z=83.085 and m/z=101.096, but we used only m/z=83.085 to evaluate its mixing ratios, because the latter contributes about 99% to the mixing ratios, represented by the three peaks. Fig. 5 below indicates elevated emission of GLVs, but a comparison of this figure with Fig. 3 in the main text suggests that there was no obviously higher excess of these GLVs from the southeast, compared to the other wounding BVOCs.

[Figure]

**Fig. 5**. *Methanol, hexanal and hexenal mixing ratios as a function of the contribution from each wind sector. The radial dimension represents the fraction of time for each wind sector during which the mixing ratios were within a certain range, as specified in the color key.*

*309-312: the fact that MEGAN does not predict local isoprene emissions is no convincing argument. Surely not all species (including invasive species) are included in the MEGAN model.*

**Answer:** We agree and have changed the text in this section as follows: "The MEGAN v2.1 simulations indicated that the known plant species in the nature park should not be a significant source of isoprene. It is possible that other local plants, such as invasive species, contributed to the observed isoprene concentration, but this

would require a large area covered by high-isoprene-emitting species to result in the observed isoprene concentration at this site." (lines 336-340).

Later in this section we mention that: "The elevated mixing ratios of isoprene+MBO from the west may be primarily attributed to the emission of isoprene from marine organisms, as discussed in Sect. 3.2.1." (lines 348-350). The discussion in Sect. 3.2.1 indicates that the Mediterranean Sea is the dominant isoprene source, but in the revised version, we emphasize the potential role of other sources on the coastline: "Yet, while it is likely that the Mediterranean Sea is the dominant isoprene source, rather than the aquaculture farms or the nature park, additional measurements on the coastline are required to quantify the contribution of other isoprene sources." (lines 458-461).

*315: I cannot follow all details of the kinetic analysis in the supplement, but I doubt that this can rule out the possibility of local emissions. I do not understand why the authors do not process their data with the eddy covariance technique. This would give a clear answer on whether there are local emissions or not.*

**Answer:**

We provide a more detailed explanation of the kinetic analysis (see Sect. S4 in the Supplement). We have also revised the original text on line 315 as follows: "However, kinetic analysis indicated that the isoprene emission from the memorial garden is much too small to account for the observed MVK+MACR associated with transported air masses from the memorial garden (see Sect. S4)." (lines 345-348).

*324: I do not agree that this has been demonstrated...*

**Answer:**

We have added a table to the Supplement that supports this (Table S3; see Table 1 below) and refer the reader to this table (lines 360-363).

**Table 1.** *Coefficients of variation for the daily mean mixing ratios of the investigated VOCs*

| VOC | Standard deviation of the daily mean mixing ratios (ppbv) | Coefficient of variation |
|---|---|---|
| Isoprene | 2.914 | 1.138 |
| Acetaldehyde | 1.539 | 0.549 |
| Acetone | 1.339 | 0.321 |
| Monoterpene | 0.227 | 0.908 |
| MVK+MACR | 0.130 | 0.650 |
| DMS | 0.014 | 0.323 |

*327: Figure S12: it would be informative to see how the scatter plot looks like for other VOCs.*

**Answer:**

In the revised Supplement (Fig. S10), we now include a scatter plot of the other BVOCs vs. temperature (see Fig. 6 below). The figure suggests that a correlation between isoprene and temperature may be masked by the high day-to-day variations in isoprene+MBO. The text has been changed accordingly: "These day-to-day variations apparently masked the seasonal correlation of isoprene with temperature (see Fig. S10)." (lines 363-364)

[Figure]

***Fig. 6**. BVOC mixing ratios versus temperature. Filled circles represent mixing ratios; exponential fit lines are in black.*

*328: Figure S10 shows data for one day. This is insufficient to make general claims.*

**Answer:**

We have added a series of figures (Figs. S12–S17) to the Supplement which clearly demonstrate that the diurnal profiles of MT and isoprene were significantly different from those of benzene, toluene and acetonitrile, which were used as proxies for transportation emissions. Note that as discussed in Sect. S3 (lines 191-192), a correlation of acetonitrile with acetone and acetaldehyde suggests that acetonitrile is probably also emitted from a biogenic source. This was most clear for 13-14 August (see Fig. 1 above).

*334: Fig 5 caption: There is no "regression" you just show a scatter plot temp vs iso+MBO*

**Answer:**

The figure caption has been amended accordingly.

*369-370: Given the fact that you measured at high E/N (140 Td) I am not so sure that such high fractions of MBO are expected at 87 Th. Maybe you can prove this by showing calibration measurements.*

**Answer:**

We do not expect that an E/N of 140 Td will almost completely fragment MBO. For example, Vlasenko et al. (2009) reported 1:3 MBO fragmentation at 135 Td. However, the reviewer makes the excellent and important point that high m87/m69 ratios are not possible if only MBO is an expected contributor to m87. This comment helped us realize that there was excess MBO signal from nonfragmenting $C_5H_{10}O$ isomers, mostly at night, that correlated better with anthropogenic sources. Thus, we have changed the label for the MBO measured at m87 MBO*, where "*" stands for methyl propyl ketone (MPK), pentanal, and other $C_5H_{10}O$ isomers. We have also updated the discussion accordingly. MBO contributions from m87 are expected in homogeneous coniferous ecosystems, but isomeric contributors are expected to m87 in other environments, in particular those with anthropogenic influences. Although MPK emissions have previously been reported from tobacco plants, the wind-sector analysis of nighttime influences on the m87 (m87/m69 > 0.3) signal indicates that the MBO excess was indeed mostly from an anthropogenic source.

*445-447: I think that it would be interesting to estimate daytime isoprene production from these lifetime values.*

**Answer:**

Daytime and nighttime isoprene production mixing ratios were calculated, assuming a lifetime of 37 min and 3.8 h during the daytime and at night, respectively. We have

added a discussion of this analysis to Sect. 3.2.1: "A rough estimation of isoprene production rate can be calculated by subtracting the isoprene loss rate, evaluated from its calculated lifetime, from its measured mixing ratios. These simplified calculations indicate a daytime and nighttime isoprene production rate ranging between $\sim 4.9 \cdot 10^{-5}$ and $1.7 \cdot 10^{-2}$ ppbv $\cdot$ s$^{-1}$ (average $5.2 \cdot 10^{-3} \pm 5.6 \cdot 10^{-3}$ ppbv $\cdot$ s$^{-1}$) and between $-1.3 \cdot 10^{-3}$ and $1.3 \cdot 10^{-3}$ (average $-1.6 \cdot 10^{-6}$ ppbv $\cdot$ s$^{-1}$ $\pm 1.4 \cdot 10^{-5}$ ppbv $\cdot$ s$^{-1}$), supporting a much smaller isoprene production rate during the night vs. daytime" (lines 476-482).

*475-478: what was the sea surface temperature in 1995 as compared to 2015?*

**Answer:**

According to Ganor et al. (2000), the sea surface temperature on August 15 and 16 when the DMS samplings were performed was 28.6°C, while between August 10 and 20, it reached up to 30.1°C-30.7°C. Accordingly, we revised the original text on lines 475-478 as follows: "Interestingly, the mixing ratios measured in this study are lower by about 1-2 orders of magnitude than those measured in the same region during August 1995 (Ganor et al., 2000). This could be attributed to a change in the marine biota as a consequence of seawater warming, considering that reported SST during mid-August 2015 (IOLR, 2015) was higher than the SST reported by Ganor et al. (2000) by up to 1.5–2.1°C" (lines 512-517).

[revised manuscript text omitted]

Overall, the day-to-day trend in the BVOC mixing ratios appears to follow the temperature, but exhibits only a relatively weak correlation with daily temperature variation (Fig. 2). DMS showed the strongest correlation with the average daytime temperature ($r^2$=0.27; see Sect. 3.2.2), corresponding to a significant increase in the mixing ratios between early summer (0.072±0.005 ppb, day of year (DOY) 188) and the end of summer (0.19±0.040 ppb, DOY 254), which decreased during the autumn (0.17±0.015 ppb, DOY 255 to 0.066±0.011 ppb, DOY 283). The other BVOCs, except for isoprene+MBO, showed a gradual increase in their average mixing ratios during the summer and early autumn (DOY 198-269; acetone from 3.74±0.767 ppbv to 4.33±0.471 ppbv, acetaldehyde from 1.64±0.595 ppbv to 3.09±0.496 ppbv, MT from 0.089±0.021 ppbv to 0.237±0.120 ppbv, MVK+MACR from 0.125±0.048 ppbv to 0.252±0.070 ppbv), and lower average mixing ratios in the autumn and early winter (DOY 270-286; DMS 0.091±0.026 ppbv, acetone 3.96±1.04 ppbv, acetaldehyde 1.86±0.97 ppbv, MT 0.139±0.064 ppbv, isoprene+MBO 0.182±0.093 ppbv, MVK+MACR 0.153±0.098 ppbv), which can be explained by the correlation with

air temperature (Fig. 2). During DOY 257-260, BVOCs showed elevated mixing ratios (daytime averages for DMS, acetone, acetaldehyde, $H_2S$, MT, isoprene+MBO and MVK+MACR were 0.122±0.016 ppbv, 13.6±3.26 ppbv, 8,140±1.18 ppbv, 0.046±0.021 ppbv, 1.97±0.215 ppbv, 7.68±0.218 ppbv and 0.644±0.084 ppbv, respectively), as well as irregular diurnal shape, which may be attributed to synoptic-scale induced processes (see Sect. S6). We therefore did not use these measurements for further analyses.

The diurnal profile of isoprene+MBO suggests a predominantly biogenic source due to a clear daytime increase and a correlation with temperature for most of the periods (Fig. 4, Figs. S3-S9). However, its day-to-day mixing ratios showed higher variability (Fig. 2), which was quite different from both DMS and the other BVOCs. The origin of the BVOCs is explored in the next section.

[Figure]

**Figure 3̶2.** The daytime average of selected VOCs. Yellow bars indicate the average daily temperature. DOY indicates the day of the year. For average diurnal profiles, see Fig. S̶5̶S3 -S̶9̶S9.

**3.2 *Origin of the BVOCs**

To explore the potential sources of the BVOCs, we calculated for each wind sector the percentage of time corresponding with several mixing-ratio ranges, individually for each species (Fig. 4̶3). Our findings indicate elevated mixing ratios for westerly and southeasterly wind components. The relatively elevated mixing ratios from the southeast can be attributed to emissions from the memorial garden, where frequent thinning of the vegetation can contribute to the generally elevated mixing ratios of plant-wounding BVOCs  which may include acetaldehyde, MVK, MACR, acetone, MT  (e.g., Brilli et al., 2011, 2012; Goldstein et al., 2004; Ormeño et al., 2011; Portillo-Estrada et al., 2015) and possibly isoprene (e.g., Kanagendran et al., 2018). While methanol, hexanal and hexenal measurements also indicated elevated mixing ratios from the southeast, our analysis

did not clearly indicate higher excess of these green-leaved species from the southeast, compared to the other wounding BVOCs (Sect. S7). The elevated mixing ratios from the west may point to an additional contribution from marine origin, such as the Mediterranean Sea and/or the aquaculture farms, considering that the measurement site is surrounded by nearly homogeneous vegetation in all directions except for the memorial garden (Fig. 1). We found a smaller relative contribution of DMS from the southeast compared to the other BVOCs. The MEGAN v2.1 simulations indicated that the known plant species in the nature park should not be a significant source of isoprene. It is possible that other local plants, such as invasive species, contributed to the observed isoprene concentration, but this would require a large area covered by high-isoprene-emitting species to result in the observed isoprene concentration at this site.

The relatively strong contribution of isoprene+MBO from the southeast can be attributed to MBO emissions from conifer trees (Gray et al., 2003) in the memorial garden. Similar trends in the day-to-day variation of MVK+MACR, isoprene oxidation products, and isoprene+MBO (Fig. 2) could imply the contribution of the memorial garden to isoprene emission. However, kinetic analysis indicated that the isoprene emission from the memorial garden is much too small to account for the observed MVK+MACR associated with transported air masses from the memorial garden (see Sect. S4). The elevated mixing ratios of isoprene+MBO from the west may be primarily attributed to the emission of isoprene from marine organisms, as discussed in Sect. 3.2.1. The origin of DMS is further addressed in Sect. 3.2.2.

**Monoterpene emission source direction (%)**

[Figure]

| | |
|---|---|
| ■ red | 0.60-1.36 ppb |
| ■ yellow | 0.35-0.60 ppb |
| ■ green | 0.15-0.35 ppb |
| ■ blue | 0-0.15 ppb |

**Isoprene+MBO emission source direction (%)**

[Figure]

| | |
|---|---|
| ■ red | 6-9 ppb |
| ■ yellow | 5-6 ppb |
| ■ green | 2-5 ppb |
| ■ blue | 0-2 ppb |

**DMS emission source direction (%)**

[Figure]

| | |
|---|---|
| ■ red | 0.115-0.296 ppb |
| ■ yellow | 0.095-0.115 ppb |
| ■ green | 0.075-0.095 ppb |
| ■ blue | 0-0.075 ppb |

**Acetaldehyde emission source direction (%)**

[Figure]

| | |
|---|---|
| ■ red | 4.5-8.5 ppb |
| ■ yellow | 3-4.5 ppb |
| ■ green | 1.5-3 ppb |
| ■ blue | 0-1.5 ppb |

**Acetone emission source direction (%)**

[Figure]

| | |
|---|---|
| ■ red | 5.5-11.6 ppb |
| ■ yellow | 4.5-5.5 ppb |
| ■ green | 4-4.5 ppb |
| ■ blue | 0-4 ppb |

**MVK+MACR emission source direction (%)**

[Figure]

| | |
|---|---|
| ■ red | 0.40-0.82 ppb |
| ■ yellow | 0.25-0.40 ppb |
| ■ green | 0.10-0.25 ppb |
| ■ blue | 0-0.10 ppb |

[Figure]

**Figure 4**. BVOC mixing ratios as a function of the contribution from each wind sector during the daytime. The radial dimension represents the fraction of time, for each wind sector, during which the mixing ratios were within a certain range, as specified in the color key.

*3.2.1 Origin apportionment of measured isoprene+MBO*

**Potential anthropogenic emission sources of isoprene+MBO**: The indication from the MEGAN v2.1 simulations that the known plant species in the nature park are not a significant source of isoprene, may suggest a significant contribution of the measured isoprene from anthropogenic sources. Moreover, as demonstrated in Sect. 3.1, the isoprene+MBO day-to-day variations differed from those of most of the other BVOCs, with remarkably high variations in its mixing ratios, ranging from 0.03 ppbv to nearly 9 ppbv (Fig. 2; Table S3 ). These day-to-day variations apparently masked the seasonal correlation of isoprene with  temperature (see Fig. S10). Two highways to the west (Fig. 1) are the major potential anthropogenic isoprene-emission sources at the site. The low correlation between the diurnal profile of isoprene and those of acetonitrile, benzene, toluene, and carbon monoxide (see Figs. S11–S17 and S18) strongly supports no significant contribution to isoprene mixing ratios from traffic on the two highways , considering that CO benzene, toluene and carbon monoxide can be used as indicator for emission from transportation. The  dominant contribution of biogenic over anthropogenic sources to isoprene is further discussed in the following.

**Potential biogenic emission sources of isoprene+MBO**: Figure 4 presents a scatter plot of isoprene+MBO mixing ratios vs. T for the six measurement periods. For the

two periods with high and low isoprene+MBO mixing ratios, there was a clear typical biogenic diurnal trend, with a maximum around noontime. This finding reinforces the notion that isoprene+MBO originates predominantly from biogenic sources. We did not, however, observe a positive correlation between isoprene+MBO mixing ratios and air $T$ in all six periods (Table 1). Furthermore, in most cases, we found no exponential increase in isoprene+MBO with air $T$, as is expected in the case of a nearby local biogenic source (e.g., Bouvier-Brown et al., 2009; Fares et al., 2009, 2010, 2012; Goldstein et al., 2004; Guenther et al., 1993; Kurpius and Goldstein, 2003; Richards et al., 2013). This might be related to the fact that the m69 signal is affected by the mixing ratios of both isoprene and MBO emitted locally and further away, while the local air temperature did not reflect changes of more distant leaf temperatures or SSTs. Therefore,  we partitioned the isoprene+MBO signal.

[Figure]

**Figure 5̶4.** Isoprene+MBO (m69) diurnal average mixing ratios and time series. (A-F) Regression scatter of  measured MBO+isoprene (ISP+MBO) vs.  temperature (upper panels) and the time series of isoprene+MBO (lower panels) for the six measurement periods: DOY 187–188 (A), DOY 197–199 (B), DOY 205–207 (C), DOY 225–226 (D), DOY 268–269 (E), DOY 282–285 (F). The  scatter  of the measured MBO+isoprene vs.  temperature (upper panels) excludes measurements associated with wind direction from the memorial garden (90°–150°).

**Partitioning of isoprene+MBO signal**: We used the fact that MBO can  also be detected at m/z=87.0805 (m87), which typically accounts for 13-25% of the total

MBO signal (Kaser et al., 2013; Park et al., 2013a, 2012, 2013b) to learn about the ratio between the isoprene and MBO mixing ratios. Note that other species, in addition to MBO, including methyl propyl ketone, pentanal, and other $C_5H_{10}O$ compounds may contribute to the m87 signal. Hence, we refer in the following to  m/z=87 as MBO* to reflect this fact. Figure 5aA presents the mixing ratios for m69 vs. m87/m69. Periods with high mixing ratios for m69 were associated with a very low m87/m69 ratio (less than 2%), which suggests that the emissions are predominantly of isoprene. Fig.5aA also indicates  that m87/m69>25% was mostly measured during the nighttime, twilight and early morning. typical~~ ratio, m87/m69, which ranges between 13-~ and 25% or higher (Fig. 5aA).

**Isoprene origin**: Figure 6 further presents the diurnal profile for m87/m69<13%, as well as the corresponding mixing ratios versus temperature, separately for each measurement day. Interestingly, some of the measurement days presented in Fig. 4 were associated with no m87/m69<13%, which is why there are fewer measurement days in Fig. 6 than in Fig. 4. The diurnal profiles in Fig. 6 support a biogenic origin for isoprene, although they were more scattered for 25-27 July. Fig.6 also demonstrates the positive correlation between the isoprene mixing ratio and  temperature during all measurement days, while on several days, a sharp increase in

isoprene with  temperature occurred for $T>\sim26\text{-}28°C$ (e.g., 6,7 July and 16 August). In general, a higher correlation with temperature was obtained for m87/m69<13% (Fig. 6) than for all m69 signals (i.e., Fig.  6 vs. Fig. 4), reinforcing the biogenic origin  of isoprene with a relatively strong dependency on temperature. Furthermore, regression of m87/m69>13% with  temperature does not indicate a clear dependency of mixing ratios on temperature, suggesting different emission controls for the MBO* and isoprene (see Fig. S21). While  the MBO* mixing ratios tended to be controlled by both  temperature and solar radiation,  isoprene was predominantly governed by the former, in agreement with a previous study (see Kaser et al., 2013).

To study the origin of isoprene, we analyzed the fraction of time for which m87/m69<13% vs. wind direction (Fig. 5b). We found that m87/m69<13% predominantly corresponds with a western origin. These results suggest a significant contribution of isoprene from the sea or the aquaculture farm located  to the west of the measurement site (Fig. 1), considering that the measurement site is nearly homogeneously surrounded by mixed-Mediterranean vegetation, except for the memorial garden to the southeast. Furthermore, MEGAN_v2.1 simulations predicted a negligible emission rate for isoprene from the nature park. In addition, the relatively high day-to-day variation in isoprene mixing ratios (Fig. 6) further support emission induced by marine organisms.

In some cases (~4% of the time), elevated m87/m69<13% was also recorded  from the southwest and northwest, which according to simulations by HYSPLIT, can be entirely attributed to transport from either the sea or the aquaculture farms (see Fig. S20). The relatively small fraction of time  during which m87/m69<13%  was from the southeast can be attributed to the emission of isoprene, while most of the

elevated isoprene+MBO from this direction (Fig. 3), can be attributed to MBO from conifers..

[Figure]

**Figure 65**.  Isoprene and MBO* origins. (aA) Scatter plot of m69 mixing ratios as a function of the m87/m69 ratio. Low and high ratios indicate a predominant contribution of MBO* (see definition in section 3.2.1) and isoprene, respectively. The orange dots were measured during the daytime and the dark blue dots during the nighttime. (bB) Fraction of time for each wind sector for which m87/m69 was <13%..

Two facts support isoprene+MBO predominant sea origin rather than the aquaculture farms. First, back trajectories using HYSPLIT show no lower mixing ratios for m87/m69<2%, isoprene+MBO also in cases when the air masses were transported from the sea but not over the aquaculture farms compared to transport of air masses over the aquaculture (e.g., Fig. 5 4 and Fig. S11S20). Second,  marine organisms have relatively short life cycles, typically a few days (Tyrrell, 2001), and would likely have a variable source strength from the aquaculture farms, which would not explain coincide with the similar measured m87/m69<2% isoprene+MBO mixing ratios for different wind directions during a specific day. Our measurements indicated no

dependence of high m87/m69<2%  mixing ratios on wind direction during the day, reinforcing the sea's dominant role in isoprene emission, rather than the aquaculture farms. Yet, while it is likely that the Mediterranean Sea is the dominant isoprene source, rather than the aquaculture farms or the nature park, additional measurements on the coastline are required to quantify the contribution of other isoprene sources.

Interestingly, the isoprene mixing ratios during the nighttime remained relatively high (~5–6 ppb) (Fig. 5aA), possibly due to relatively small oxidative sink strength during the night. The daytime and nighttime isoprene lifetime can be estimated based on its reaction with OH, $NO_3$ and $O_3$. We estimated the average daytime OH and nighttime $NO_3$ concentrations, based on the MINOS campaign in Finokalia, Crete (Berresheim et al., 2003; Vrekoussis et al., 2004), at $4.5 \cdot 10^6 \frac{molec}{cm^3}$ (Berresheim et al., 2003), and $1.1 \cdot 10^8 \frac{molec}{cm^3}$ (Vrekoussis et al., 2004), respectively. Using these concentrations, the reported rate constants for isoprene with OH and $NO_3$ of $1 \cdot 10^{-10} \frac{cm^3}{molec \cdot s}$ (Stevens et al., 1999) and $5.8 \cdot 10^{-13} \frac{cm^3}{molec \cdot sec}$ (Winer et al., 1984), respectively, and measured $O_3$ levels, we obtained daytime and nighttime isoprene lifetimes of ~37 min and ~3.8 h, respectively. Considering the relatively moderate decrease in the measured isoprene during the night (Figs. S11–S16), this result indicates stronger isoprene emissions  during the daytime, but does not rule out nighttime isoprene emissions.

A rough estimation of isoprene production rate can be calculated by subtracting the isoprene loss rate, evaluated from its calculated lifetime, from its measured mixing ratios. These simplified calculations indicate a daytime and nighttime isoprene production rate ranging between ~$4.9 \cdot 10^{-5}$ and $1.7 \cdot 10^{-2}$ ppbv ·

$s^{-1}$ (average $5.2 \cdot 10^{-3} \pm 5.6 \cdot 10^{-3}$ ppbv $\cdot$ s$^{-1}$) and between $-1.3 \cdot 10^{-3}$ and $1.3 \cdot 10^{-3}$ (averaged $-1.6 \cdot 10^{-6}$ppbv $\cdot$ s$^{-1}$ $\pm 1.4 \cdot 10^{-5}$ ppbv $\cdot$ s$^{-1}$), supporting a much smaller isoprene production rate during the night vs. daytime.

[Figure]

**Figure 76**. Isoprene (m87/m969<13%)  diurnal average mixing- ratio  and dependence

on temperature. Upper panels show  scatter plot between measured m87/m96<13% and

temperature, as well as the corresponding regression equation and nonlinear exponential coefficient

($R^2$) in cases when $R^2 > 0.50$. Lower panels present that of m87/m96<13%. Yellow shaded area represents daylight hours.

*3.2.2 Origin and characterization of DMS emission*

The discussion in Section 3.1 suggests that DMS is primarily emitted from the west, pointing to a dominant marine emission source, with the less elevated mixing ratios probably associated with emission from vegetation. According to the MEGAN v2.1 simulation, the natural park's vegetation is a potent source of DMS (average flux=0.477 $\frac{mg}{m^2 \cdot hr}$), slightly higher than the flux measured from insolated branches (Jardine et al., 2015; Yonemura et al., 2005), while our analysis points to a stronger emission from the memorial garden (see Fig. 3). As with isoprene, insensitivity of DMS mixing ratios to wind direction, for westerly winds, rules out a significant contribution of the aquaculture farms to the measured DMS. This suggests that the sea is a major source for DMS, with an apparently strong dependency on temperature (Figs. 2, 3). DMS showed much less day-to-day variations in its mixing ratios compared to isoprene and other BVOCs. This corresponded with a clear day-to-day correlation of DMS mixing ratios with air temperature (Fig. 2). Figure 7 demonstrates a clear increase in the mixing ratios with air temperature, throughout the measurement period. Note that no significant dependency of DMS on global solar radiation was observed.

The DMS mixing ratios peaked at ~0.18 ppbv. This figure is about an order of magnitude lower than at the ocean surface (Tanimoto et al., 2014), about an order of magnitude lower than in the Southern Ocean (Koga et al., 2014), slightly lower than the maximum concentrations in the south Indian Ocean (Aumont et al., 2010), and similar to the maximum concentrations on the coasts of Tasmania (Aumont et al.,

2010). Interestingly, the mixing ratios measured in this study are lower by about 1-2 orders of magnitude than those measured in the same region during August 1995 (Ganor et al., 2000). This  could be attributed to a change in the marine biota as a consequence of seawater warming, considering that reported SST during mid-August 2015 (IOLR, 2015) was higher than the SST reported by Ganor et al. (2000) by up to 1.5–2.1°C.

[Figure]

**Figure 87.** Daytime DMS mixing ratios from the western sector (marine source) as a function of the temperature along the measurement campaign. An exponential fit between the two is included.

**3.2.3 Origin and characterization of other BVOCs**

Our findings in Figure 3 2 strongly suggest a common source for other BVOCs with isoprene. We could not, however, use a wind-direction-based analysis to indicate BVOCs' origin from the sea, since both sea and vegetation are located to the west of the measurement point (see Fig. 1), and in contrast to isoprene, the other BVOCs were indicated by MEGAN v2.1 to be locally emitted. Furthermore, those BVOCs were less variable with wind direction than was isoprene. We used MT summer measurements from two other sites in Israel to assess whether MTs are likely to be transported to the measurement site.

We used the ratio between MT flux and mixing ratio at the three sites as a basis to address this inquiry. Note that according to the MEGAN v2.1 simulations (see Sect. 2.3), the MT emissions in Ramat Hanadiv were driven by *Quercus calliprinos* (48.1%), *Pistacia lentiscus* (19.8%), *Phillyrea latifolia* (7.12%) and *Cupressus* spp. (6.17%), as well as other species (see Sect. S5), in contrast to the two *Pinus halepensis* plantations, Birya and Yatir. While the fact that MT is not emitted by the same vegetation species should not significantly affect our analysis, we recognize that

there may be differences in the MT composition and atmospheric oxidation capacity at the three sites which would influence MT lifetimes and lead to some differences in the flux--to--concentration ratios. According to MEGAN_v2.1, 
[revised manuscript text omitted]

Yuan, B., Koss, A. R., Warneke, C., Coggon, M., Sekimoto, K. and De Gouw, J. A.: Proton-Transfer-Reaction Mass Spectrometry: Applications in Atmospheric Sciences, Chem. Rev., 117(21), 13187–13229, doi:10.1021/acs.chemrev.7b00325, 2017.